



**Formation and origin of Fe-Si oxyhydroxide deposits at the ultra-slow spreading**
**Southwest Indian Ridge**
**Authors:**
Kaiwen Ta[1, 2], Zijun Wu[1*], Xiaotong Peng[2] and Zhaofu Luan[1]
[1]School of Ocean and Earth Science and State Key Laboratory of Marine Geology,
Tongji University, Shanghai, China,
[2]Deep Sea Science Division, Institute of Deep Sea Science and Engineering, Chinese
Academy of Sciences, Sanya, China.
**Corresponding author:**
Zijun Wu, (wuzj@tongji.edu.cn)
**Abstract**
Low-temperature hydrothermal system is dominated by Fe-Si oxyhydroxide
deposits. However, the formation process and mechanism on modern hydrothermal
Fe-Si oxyhydroxides at ultra-slow spreading centers remain poorly understood. The
investigation presented in this paper focuses on six Fe-Si deposits collected from
different sites at the Southwest Indian Ridge (SWIR). The mineralogical and
geochemical evidence showed significant characteristics of a low-temperature
hydrothermal origin. The Mössbauer spectra and iron speciation data further provided



an insight into iron-bearing phases in all deposits. Two different types of
biomineralized forms were discovered in these deposits by Scanning Electron
Microscopy analysis. Energy-dispersive X-ray spectrometry and nano secondary ion
mass spectrometry revealed that distinct biogenic structures were mainly composed of
Fe, Si, and O, together with some trace elements. The Sr and Nd isotope compositions
of Fe-Si deposits at the SWIR were closely related to interaction between
hydrothermal fluids and seawater. The remarkably homogeneous Pb isotope
compositions can be attributed to hydrothermal circulation. Based on these findings,
we suggest that microbial activity plays a significant role in the formation of Fe-Si
oxyhydroxides at the at ultra-slow spreading SWIR. Biogenic Fe-Si oxyhydroxides
potentially provide insights into the origin and evolution of life in the geologic record.
**1 Introduction**
Hydrothermal Fe-Si-oxyhydroxide deposits are widespread in many geological
settings, such as mid-ocean ridges (Alt, 1988; Benjamin et al., 2006; Dekov et al.,
2010; Peng et al., 2015), back-arc spreading centers (Iizasa et al,. 1998; Hein et al.,
2008; Sun et al., 2012), seamounts (Karl et al., 1989; Boyd and Scott, 2001; Emerson
and Moyer, 2002; Singer et al., 2011), and intra-plate submarine volcanoes (Edwards
et al., 2011; Fleming et al., 2013). Fe-Si oxyhydroxides in the form of yellowish to
brown chimneys, mounds and flat-lying deposits have often been observed in
low-temperature hydrothermal fields (Emerson and Moyer, 2002; Peng et al., 2015;
Johannessen et al., 2016; Ta et al., 2017). In general, modern low-temperature



hydrothermal systems are the product of diffuse hydrothermal fluids and/or the
conductive cooling of high-temperature hydrothermal fluids mixed with seawater
(German et al., 1990). A pronounced excess of ferrous iron and dissolved silica are
typical characteristics of hydrothermal vent fluids (Tivey, 2007). Low-temperature
hydrothermal Fe-Si oxyhydroxides are considered as important hydrothermal products
which reflect the diffusion and evolution of hydrothermal fluids. The mineralogical
and geochemical compositions of such deposits have provided insight into their
formation mechanisms (Dekov et al., 2010; Sun et al., 2015; Ta et al., 2017). Recently,
studies focused on the origin of Fe-Si-oxyhydroxides have been increasing, however
little is currently known about the links between these deposits and microbial activity
at the ultra-slow spreading Southwest Indian Ridge (SWIR).

The ultra-slow spreading SWIR represents the longest segment of the world's

slowest-spreading ridge (German et al., 2010; Husson et al., 2015). The SWIR is
characterized by a spreading rate of approximately 12–15 mm/yr, a lack of transform
faults, and extensive exposures of mantle peridotites (Dick et al., 2003; Niu et al.,
2015). Compared to fast-spreading ridge systems, hydrothermal activity along the
ultra-slow spreading SWIR may have greater chemical and thermal fluxes (German et
al., 2010). However, the composition and depth of oceanic crust at the SWIR seems to
be different in many respects from average oceanic crust, and the data suggest the
presence of thickened crust or a large thermal anomaly in the region (Sauter et al.,
2009; Zhang et al., 2013; Niu et al. 2015). The diversity in styles of hydrothermal
vents at the SWIR has been of particular interest, particularly the activity of





high-temperature and low-temperature vents (German et al., 1998; Tao et al., 2012).
Isotope analyses of mid-ocean ridge basalts have revealed a clear distinction between
the Southwest Indian ridge and the Pacific and Atlantic ridge compositions (Hamelin
and Allègre, 1985; Vlastélic et al., 1999). Previous studies have confirmed that
plume-ridge interactions have produced geochemical and geophysical anomalies
along the Indomed and Gallieni fracture zones beneath the SWIR (Breton et al., 2013;
Yang et al., 2017). Recently, increasing attention has been paid to the petrology,
element geochemistry, microbial communities and biogeography of the SWIR (Li et
al., 2015; Chen et al., 2016; Ji et al., 2017; Zhou et al., 2018; Zhang et al., 2018).
Although microbial activity was revealed to play an important role in the formation of
Fe and Si minerals in low-temperature hydrothermal fields (Dekov et al., 2010), there
are a few studies that show the presence of biomineralized structures encrusted by
Fe-Si oxyhydroxides in the SWIR hydrothermal systems, indicating the oxyhydroxide
deposits may also be of biogenic origin (Peng et al., 2011; Sun et al., 2015).

Here, we report the geochemical and geomicrobiological characterization of

Fe-Si deposits from the ultra-slow spreading SWIR. Scanning electron microscope
(SEM), X-ray diffraction (XRD), inductively coupled plasma-mass spectrometry
(ICP-MS), nano secondary ion mass spectrometry (nanoSIMS), Pb-Sr-Nd-O isotopic
analysis, Mössbauer Spectroscopy, and sequential iron mineral extraction experiments
were used to investigate: (1) the geochemical and morphological characteristics of
biogenic Fe-Si oxyhydroxides, (2) the role of microbial activity in the formation of



the Fe-Si oxyhydroxides, (3) the implications of the Sr-Nd-Pb isotopic content of the
low-temperature Fe-Si deposits at the ultra-slow spreading SWIR.
**2 Geological Setting**
This study is focused on the ultra-slow spreading Southwest Indian ridge
segments 27 and 28, as defined by Cannat et al. (1999), incorporating the Indomed
and Gallieni Fracture Zones (46.0 °E to 52.0 °E), where the central shallow section of
the SWIR has an overall 15° obliquity. Geophysical and geochemical data have
shown that this area is a V-shaped shallow domain associated with thickened crust and
robust magmatism (Lin and Zhang., 2006). Tectonic and volcanic processes result in
the growth of the oceanic crust at the SWIR (Niu et al. 2015; Li et al., 2015). Previous
geological surveys have indicated that the robust melt supply may be associated with
the Marion and Crozet hotspots (Georgen et al., 2001; Sauter et al., 2009). A study by
Yu et al. (2018) showed that segments 27 and 28 of the ridge had contrasting tectonic
and magmatic processes. An inactive hydrothermal field was discovered near the
center of segment 27 (Zhao et al., 2013). The axial depth of this segment has become
shallower and experienced a dramatic increase in magma supply since 8–11 Ma
(Sauter et al., 2009). However, the Longqi hydrothermal field of segment 28, at
49.6 °E, has been confirmed to remain active, at a water depth of around 2750 m (Tao
et al., 2012).
**3 Materials and Methods**
**3.1 Sample Collection and Sample Descriptions**





110   Fragile and porous hydrothermal deposits (samples 20V-T8, 21V-T1, 21V-T7

111  and 33II-T2) were collected by a TV grabber during a cruise of the R/V DaYang

112  YiHao conducted by the China Ocean Mineral Resource R&D Association (COMRA)

113  at the SWIR, from 2008 to 2015. The yellowish 21V-T1 and brown 21V-T7 samples

114  were located ~20 km north of the SWIR, and the brown 20V-T8 sample was located

115  ~8 km to the north. The purple-red 34II-T22 sample was collected from the axis of the

116  SWIR. Sample DIV95 was recovered from the Longqi field by the Human Occupied

117  Vehicle (HOV) 'Jiaolong' diving cruise in 2015. This sample was very friable with a

118  layered structure. The layers were nearly parallel and displayed an obvious color

119  change (Fig. 2a). The upper layer of DIV95 was thin (~2–3 cm), and composed of

120  orange-yellowish Fe-Si oxyhydroxides, whereas the bottom portion comprised a

121  thicker (~3–5 cm), black layer of mixed Fe-Si oxyhydroxides and Mn-oxides. After

122  recovery, the fresh samples were immediately divided into subsamples for

123  mineralogical, geochemical and microscopic analyses. A small amount of the

124  subsamples to be used for SEM analysis were fixed with 8% formaldehyde at −20 °C

125  in sterile bags, and a separate amount was stored at 4 °C prior to nanoSIMS analysis.

126  The rest of the samples were stored in anoxic hermetic bags at −20 °C to avoid

127  oxidation.

128  **3.2 Analytical Methods**

129  **3.2.1 Bulk Chemistry**

130    Chemical compositions of the samples were determined by X-ray fluorescence

131  (XRF) spectrometry and ICP-MS. Major elements were measured using XRF





spectrometry (Shimadzu XRF-1800) with operating conditions of 40 kV and 95 mA.
Six samples were powdered to 200 mesh size for major elements analysis. Powdered
samples were then leached twice in 6 M HCl for 2 hours at 100 °C, followed by
ultrasonic leaching in Milli-Q water. Major elements were analyzed quantitatively
after the fusion of 0.1 g of sample material with 3.6 g of dilithium tetraborate at
1050 °C for 16 min. Trace and rare earth element compositions of the samples were
determined by ICP-MS using a Thermo VG-X7 mass spectrometer. Samples were
dissolved using a solution of $HNO_3$ + HF on a hot plate. The eluted samples were
diluted by 2% $HNO_3$ for trace element quantification (Peng et al., 2011). The
precision determined from sample duplicates, as well as from repeated analyses, was
better than 5%.

### 3.2.2 Mineralogy

XRD was employed to characterize the mineralogy of the particles of interest.

Samples were freeze dried under anoxic conditions to avoid oxidation during drying.
The subsamples were thoroughly ground using a pestle and a mortar, followed by
analyses using a D/max2550VB3+/PC X-ray diffractometer (Rigaku Corporation)
with Cu Kα radiation at 35 kV and 30 mA. Diffraction angles corresponding to the
unique crystal structure of each mineral were measured. The scan speed was 2°
2θ/min, and the resolution was 0.02° 2θ.

### 3.2.3 Morphological Diversity

SEM was employed to determine the morphological diversity of the

hydrothermal deposits. Freeze-dried subsamples were fixed onto aluminum stubs with



two-way adherent tabs and allowed to dry overnight. Subsequently, the samples were
sputter coated with gold for 30 seconds. All samples were examined using an FEI
Apreo SEM equipped with an EDAX energy-dispersive X-ray spectrometer (EDS).
The SEM was operated at 2 kV with a working distance of 10 mm to facilitate
optimum image collection whilst minimizing charging and sample damage. For EDS
analyses, an accelerating voltage of 20 kV was used to generate sufficient X-ray
counts.
**3.2.4 Pb-Sr-Nd isotopes**
Sr, Nd, and Pb isotopic compositions were quantified in the Laboratory for
Radiogenic Isotope Geochemistry at the University of Science and Technology of
China, using a Phoenix-Thermal Ionization Mass Spectrometer (Isotopx, UK) for Sr
and Nd analysis, and an IsoProbe-Thermal Ionization Mass Spectrometer (GV
[formerly Micromass], UK) for analysis of Pb. The detailed analytical procedure for
Nd, Pb, and Sr isotopic measurements follows that described by Chen et al. (2000,
2007). Sample powders (~100 mg) used for isotopic analysis were dissolved in $HNO_3$
+ HF solution, and then transferred to a 6 M HCl solution. Pb was fixed to Ta
filaments using Si-gel. Sr was loaded onto preconditioned Ta filaments using a Ta-HF
activator. Nd was loaded as phosphate onto preconditioned Re filaments. Sr and Nd
isotopic ratios were normalized to an $^{86}Sr/^{88}Sr$ of 0.1194 and an $^{143}Nd/^{144}Nd$ of 0.7219
during runtime. Measured values for the NBS 987 Sr and La Jolla Nd standards were
$0.710265 \pm 12$ ($2\sigma$) for $^{86}Sr/^{88}Sr$, and $0.511862 \pm 10$ ($2\sigma$) for $^{143}Nd/^{144}Nd$ . The Pb
isotope data were periodically checked against NBS 981, which produced means of





$^{206}Pb/^{204}Pb = 16.9416 \pm 13 \ (2\sigma)$, $^{207}Pb/^{204}Pb = 15.500 \pm 13 \ (2\sigma)$ and $^{208}Pb/^{204}Pb =$
$36.7262 \pm 31 \ (2\sigma)$ (Baker et al., 2004). The internal precision of Pb isotope data was
estimated to be less than 0.03%.
**3.2.5 Oxygen Isotope Analysis**
Six freeze-dried samples powdered to 200 mesh size were purified using the
following procedure. Carbonate was treated using 10% (vol/vol) acetic acid by
sonication for 2 h. The Fe and Mn oxides were removed using a mixture of 1 M
hydroxylamine hydrochloride and 25% (vol/vol) acetic acid. Organic matter was
digested by adding aqua regia. The final detritus was rinsed three times with distilled
water and dried in an oven at approximately 55 °C. Stable oxygen isotope analyses
were performed using a MAT-253 mass spectrometer at the Institute of Mineral
Resources, Chinese Academy of Geological Sciences, China. Oxygen isotope data
were collected from approximately 20 mg purified samples, using $CO_2$ generated
from silicates by heating the powder with a $CO_2$ laser, using $BrF_5$ as the fluorinating
reagent (Cole et al., 2004). The resultant oxygen was converted to $CO_2$ on a
platinum-coated carbon rod. The isotopic data are reported relative to the Standard
Mean Ocean Water (SMOW) with a precision of 0.2‰.
**3.2.6 Fe behavior and oxidation state**
The $^{57}Fe$ Mössbauer spectra of six homogenized samples were recorded in
transmission geometry using a conventional constant-acceleration spectrometer. An 8
mCi activity $^{57}Co$ source supplied γ rays for the measurements. The spectra were
recorded at room temperature. The spectrometer was calibrated using a standard α-Fe





foil, and the reported isomer shifts are relative to the center of the α-Fe spectrum. The
fit of the Mössbauer spectra was evaluated using doublets of Loretzian peaks via the
least squares method, with the WinNormos-for-Igor 3.0 program. The ideal adsorber
thickness values were generated with the *Recoil* program (Lagarec and Rancourt,

1998).

### 3.2.7 Iron Speciation

Iron speciation was extracted from the deposits following the sequential

extraction technique developed by Poulton and Canfield (2005). In brief,
approximately 0.5 g of each dried SWIR sample was accurately weighed. The
samples were powdered and added to Teflon tubes. The samples were then mixed with
the appropriate solvent for a defined period of time (Table 4). Subsequently, the
samples were centrifuged at 4000 rpm. The extraction was decanted and filtered
through a 0.2 μm membrane. Between each step, the samples were washed with
distilled water. Iron concentrations in the extracts were determined using an
Inductively Coupled Plasma Atomic Emission Spectrometer (ICP-AES, Perkin Elmer
Optima 3000) with a relative standard deviation of less than 2%. The contents of all
samples were normalized to extracted dry deposits (μg/g).

### 3.2.8 Ion Distribution

NanoSIMS was employed to characterize the nanometer- to micrometer-scale

distribution of $^{12}C^-$, $^{12}C^{14}N^-$, $^{32}S^-$, $^{27}Al^{16}O^-$, $^{55}Mn^{16}O^-$ and $^{56}Fe^{16}O_2^-$ in Fe-Si deposits
that were spread on glass slides. NanoSIMS analyses were performed at the Institute
of Geology and Geophysics, Chinese Academy of Sciences, using a CAMECA





NanoSIMS 50 L (CAMECA, Paris, France). This nanoSIMS is capable of sub-50 nm
lateral resolution while imaging negatively charged secondary ions, are samples have
been sputtered with $Cs^+$ primary ions. Each region of interest was presputtered using a
150 pA beam current and an ion dose of $N > 5 \times 10^{16}$ ions/cm$^2$ (Gnaser, 2003). This
treatment removed any surface contaminants, implanted $Cs^+$ ions into the sample
matrix, and enabled an approximately steady state of ion emission to be reached.
Using a $Cs^+$ primary beam, negative secondary ions ($^{12}C^-$, $^{12}C^{14}N^-$, $^{32}S^-$, $^{27}Al^{16}O^-$,
$^{55}Mn^{16}O^-$ and $^{56}Fe^{16}O_2^-$) were sputtered from the sample surface with a beam current
of c.2.5 pA, and were detected in multicollection mode (Ta et al., 2017).
**4 Results**
**4.1 Geochemistry**
Major and trace element compositions of the Fe-Si deposits are presented in
Table 1. All analyzed deposits had $Fe_2O_3$ contents ranging from 11.56 to 64.33 wt%,
and $SiO_2$ contents ranging from 27.22 to 80.20 wt%. The highest $Fe_2O_3$ and $SiO_2$
concentrations were found in the purple-red 34II-T22 and brown 20V-T8 deposits,
respectively. The two subsamples of DIV95 showed different chemical compositions.
The $MnO_2$ concentration of the black layer of DIV95-2 was 30.13 wt%, while the
content of $SiO_2$ in DIV95-2 was lower than in DIV95-1. The deposits displayed
limited variability in their $P_2O_5$ content, which ranged from 0.149 to 0.898 wt%. The
Fe/Mn ratios of the deposits varied over a broad range from 1.41 to 723.53, and the
Al/(Al + Fe + Mn) ratios were extremely low (< 0.003).





The studied deposits contained very low amounts of the majority of trace

elements and REEs. The values of total rare earth elements (ΣREE) in the different

deposits varied from 1.135 to 18.96 ppm, with an average of 7.67 ppm. The

concentration of ΣREE in the 34II-T22 sample was the highest of all deposits. The

REE distribution patterns of the Fe-Si deposits exhibited both negative Ce and

positive Eu anomalies. The deposits showed a slight enrichment in light REE (LREE)

relative to heavy REE (HREE) (Fig. 2). Fe-Si deposits had significantly higher

large-ion lithophile element (such as Sr, U, Rb and Ba) contents than high field

strength element (such as Hf, Th, Ta and Nb) contents. The 34II-T22 sample was

noticeably enriched in trace elements such as Pb, V, Cu, Co, Ni, Zn, and U, but

depleted in Li and Ba, relative to the other samples. The $Fe_2O_3/SiO_2$ ratios of the

21V-T1 and 21V-T7 deposits showed a narrow range (0.32–0.36), however the

20V-T8 deposits had the lowest $Fe_2O_3/SiO_2$ ratio of all deposits. The Fe/REE ratios of

the deposits varied between 1.57 and 12.99. The 20V-T8 deposit, that was richer in Si

and slightly depleted in Fe compared to the 34II-T22 deposit, also had lower

compositions of the trace elements and REE relative to 34II-T22. Fe/Mn and Fe/REE

ratios were lowest in the DIV95-1 and DIV95-2 samples.

The $\delta^{18}O$ values of all the deposits ranged between 16.56‰ and 35.87‰ (Table

2). The O isotopic composition of the hydrothermal deposits was reflected in the

poorly crystalline Fe-Si oxyhydroxides, which precipitated from hydrothermal fluids.

O isotopic fractionation is often used to calculate precipitation temperature in

hydrothermal environments. We reference the $\delta^{18}O_{hydrothermal\ fluid}$ data of the Kairei





hydrothermal field in the Central Indian Ridge (Gamo et al., 2001). The precipitation
temperature was calculated by Kita et al. (1985). The calculated results indicate that
the deposits would have precipitated at temperatures across the range
28.76–114.66 °C, which supports their low temperature origin.
The Sr-Nd-Pb isotopic compositions of the deposits are presented in Table 2 and
Figure 7. The Sr isotopic compositions of the deposits showed only slight variation
($^{87}$Sr/$^{86}$Sr = 0.707937–0.709150), and had values similar to present-day seawater
($^{87}$Sr/$^{86}$Sr = 0.70917) (Burke et al., 1982). $^{143}$Nd/$^{144}$Nd values of the deposits varied
from 0.512332 to 0.512801, corresponding to εNd values ranging from -6 to 3.2.
Positive εNd values implied the presence of an SWIR upper mantle component. Pb
isotope compositions showed clear homogenization ($^{206}$Pb/$^{204}$Pb = 18.2694–18.4829,
$^{207}$Pb/$^{204}$Pb = 15.5488–15.6404, $^{208}$Pb/$^{204}$Pb=38.2145–38.4493). In contrast,
$^{207}$Pb/$^{204}$Pb and $^{208}$Pb/$^{204}$Pb ratios of the samples were found to be relatively high
compared to SWIR and SEIR basalts (Figs. 7a–b). $^{206}$Pb/$^{204}$Pb ratios of the deposits
correlated well with $^{207}$Pb/$^{204}$Pb and $^{208}$Pb/$^{204}$Pb, except for 21V-T7. The majority of
SWIR deposit compositional data were clearly distinct from mid-Pacific and Antarctic
Ridge basalts and Arctic marine sediments (Fig. 7). Additional details of the isotopic
characteristics of the deposits were shown by plots of $^{87}$Sr/$^{86}$Sr versus $^{143}$Nd/$^{144}$Nd and
$^{206}$Pb/$^{204}$Pb (Fig.s 7c–d).The samples showed significant similarity to the
low-temperature iron-silica-rich deposits from the mid-Atlantic ridge (Fig.s 7c–d).
There was no overlap between the isotopic signature of our samples and other
geological settings.



### 4.2 Mineralogy of Fe-Si deposits and Mössbauer Spectroscopy


The XRD results showed that 2-line ferrihydrite, pyrite, natrojarosite, opal and
birnessite comprised the major minerals in the samples (Fig. S2). In the spectrum of
sample 33II-T8, a broad peak centered at 4.08 Å suggested the presence of opal. The
spectral peaks from samples 21V-T1, 21V-T7 and DIV 95-1 appeared at 3.83 Å and
2.21 Å, indicating the presence of pyrite. The spectral signature of birnessite was most
clearly observed in sample DIV95-2, at d = 7.06 and 2.45 Å. A small amount of
birnessite was observed in DIV95-1, which was presumed to be caused by the residual
black layer. Poorly crystalline two-line ferrihydrite, characterized by appearance
peaks at d = 2.62 Å and 1.51 Å, was the principal mineral observed in the spectra of
sample DIV95-2. Natrojarosite was also present in DIV95-1 and 21V-T7 deposits. In
addition, halite was observed in our samples, which presumably formed by
evaporation.
$^{57}$Fe Mössbauer spectroscopy has been determined to be one of the most efficient
methods for studying the behavior and oxidation state of Fe (Murad and Schwertmann,
1980). The corresponding Mössbauer parameters and the identification of phases in
the spectra are presented in Figure 5 and Table 3. $Fe^{3+}$ occurred primarily in four-fold
and six-fold coordination with oxygen, representing $^{IV}Fe^{3+}$ and $^{VI}Fe^{3+}$ components
(Burkhard, 2000). The Mössbauer spectrum of the DIV95-1 was fitted with two
quadrupole doublets. One doublet (IS = 0.59 mm/s, QS = 0.84 mm/s) was
characteristic of octahedrally-coordinated ferrihydrite (Murad and Johnston, 1987;
Murad and Cashion, 2004). The other doublet (IS = 0.33 mm/s, QS = 0.79 mm/s) was



comparable to those of two-line ferrihydrite reported in previous studies (Murad and
Schwertmann, 1980; Johnston and Lewis, 1983; Murad and Johnston, 1987; Berquó et
al., 2007). Likewise, the Mössbauer spectrum of DIV95-2 was also fitted with two
different quadrupole doublets. One doublet (IS = 0.34 mm/s, QS = 0.55 mm/s) was
characteristic for lepidocrocite (Murad and Schwertmann, 1980; Murad, 1984). The
other doublet (IS = 0.34 mm/s, QS = 0.85 mm/s) was analogous to those of two-line
ferrihydrite reported in hydrothermal deposits (Peng et al., 2013). However, the
spectrum of the 34II-T22 was fitted with a single quadrupole doublet. Parameters of
IS (0.36 mm/s) and QS (0.72 mm/s) indicated that the doublet could be ascribed to
two-line ferrihydrite (Murad, 1988; Wade et al., 1999). Furthermore, parameters of IS
= 0.40 mm/s, QS = 0.65 mm/s and IS = 0.40 mm/s, QS = 1.11 mm/s for sample
21V-T1    were    interpreted    to    reflect    the    presence    of    goethite    and
octahedrally-coordinated ferrihydrite, respectively (Oh et al., 1998; Murad and
Schwertmann, 1980). The 21V-T7 sample showed two quadrupole doublets with IS =
0.34 mm/s, QS = 0.71 mm/s and IS = 0.56 mm/s, QS = 0.84 mm/s, corresponding to
two-line ferrihydrite and octahedrally-coordinated ferrihydrite, respectively (Oh et al.,
1998; Murad and Schwertmann, 1980). Moreover, sample 20V-T8 displays a further
type of Mössbauer spectrum, fitted using two quadrupole doublets. Values of IS =
0.40 mm/s, QS = 1.01 mm/s and IS = 0.40 mm/s, QS = 0.58 mm/s were in accordance
with those previously reported for lepidocrocite and ferrihydrite (Murad and
Schwertmann, 1980; Berquó et al., 2007).
**4.3 Sequential Iron Mineral Extraction**





Sequential extraction of iron minerals identified four iron-bearing phases, as
shown in Table 4 and Figure 6. $Fe_{Carb}$, the adsorbed iron and carbonate associated iron
pool, was the least abundant of the iron-bearing phases, and varied between 11.38
μmol/g and 28.70 μmol/g. The proportion of $Fe_{Carb}$ in the 34II-T22, 21V-T7 and
20V-T8 samples was higher than in DIV95-2 (Fig. 6). $Fe_{OX1}$, the easily reducible
ferric iron oxide pool, mainly recorded the presence of ferrihydrite and lepidocrocite.
$Fe_{OX1}$ concentrations in the DIV95-1, 34II-T22, 21V-T7 and 20V-T8 samples were all
similar, at approximately 111.71–118.48 μmol/g, with sample DIV95-2 having a
much lower content of about 0.4 umol/g. $Fe_{OX2}$, the ferric iron (hydr)oxide pool,
mainly recorded the presence of goethite and hematite. $Fe_{OX2}$ concentrations of all
deposits were very similar, varying from 218.13 to 226.51 μmol/g. The most abundant
Fe-bearing phase was $Fe_{PRS}$, the poorly reactive sheet silicate pool in Fe-Si deposits.
Sample DIV95-1 had a lower $Fe_{PRS}$ concentration than sample DIV95-2, but was still
rather high at about 647.46 μmol/g.
**4.4 Microtextures, Micromorphologies and Compositions**
SEM observations showed that different types of structures were abundant in the
hydrothermal Fe-Si deposits. Morphologies included rod-like sheaths, rosette
spherical structures, mineralized spheroids, ribbon-like helical filaments, threadlet
filaments, branched structures and twisted stalks (Fig. 4). The orange-yellowish
DIV95-1 deposit was primarily composed of mineralized rod-like forms with a
network-like structure (Fig. 4a). However, the black DIV95-2 deposit was
characterized by rosette spherical structures with a main component of Mn, according





to EDS results (Fig. 4j). Spherical morphologies encrusted by iron and silicon were
present in the yellowish 21V-T1 deposits (Fig. 4d). A wide morphological diversity of
Fe-Si filamentous forms was also identified, including twisted filaments, curved
filaments, and branched filaments. Filamentous structures were particularly abundant
in the purple-red 34II-T22 and brown 21V-T7 deposits (Figs. 4e–g). The threadlet
filaments had a diameter of 0.5 to 1 μm and were up to 100 μm in length (Fig. 4g).
These filaments resemble the Fe oxyhydroxide stalks produced by the
chemolithotrophic Fe-oxidizing bacterium *Marirpofundus ferrooxydans*. The hollow
tube observed in sample 34II-T22 was about 10–50 μm in length and 1–3 μm in
diameter (Fig. 4e), which is currently considered the characteristic trace of *Leptothrix*
*ochracea*. In addition, branched sheaths and twisted stalks were also observed in
20V-T8 (Figs. 4h–i), with abundant spheroids scattered throughout the matrix of
branched sheath structures (Fig. 4i). These mineralized sheaths and stalks are related
to the metabolism of Fe-oxidizing bacteria previously reported in deep sea
hydrothermal environments (Emerson and Moyer, 2010; Edwards et al., 2011; Peng et
al., 2015; Johannessen et al., 2016; Chan et al., 2016). EDS analyses revealed that the
branched sheaths and twisted stalks were composed of Fe, Si, and small amounts of
Mg and Ca (Fig. 4l).
**4.5 Isotopic Signals Revealed by nanoSIMS**
The results of nanoSIMS mapping of Fe-stalk coming from sample 34II-T22 are
shown in Figure 8. NanoSIMS has a higher sensitivity than SEM-EDS for most
elements. In regions of interest, a pronounced intensity of $^{56}Fe^{16}O_2$ signals was





observed in the stalk. The co-location of $^{56}Fe^{16}O_2$ and $^{27}Al^{16}O$ signals indicated that Fe
and Al may originate from hydrothermal fluids and co-precipitated with the stalk. The
elevated $^{55}Mn^{16}O$ signals were correlated with $^{56}Fe^{16}O_2$ signals, suggesting that Mn is
probably formed through the adsorption of Mn onto the Fe-stalk. $^{12}C$ and $^{12}C^{14}N$
intensities are known to be very sensitive to biologically-derived materials (Herrmann
et al., 2007). Therefore, the entire stalk was expected to show a high concentration of
C and N elements, but $^{12}C$ and $^{12}C^{14}N$ signals were relatively low from this stalk.
Although we observed more $^{12}C^{14}N$ signals than $^{12}C$ signals in the Fe-stalk, the yields
of $^{12}C^{14}N$ secondary ions adjacent to the surrounding material were much higher.
**5 Discussion**
**5.1 Origin of Fe-Si oxyhydroxide deposits at the SWIR**
The Fe-Si oxyhydroxide deposits from the SWIR show mineralogy and chemical
composition similar to those from other tectonic settings that have been interpreted to
be of low temperature hydrothermal origin. The Fe-Si oxyhydroxides are
characterized by enriched Fe and Si, along with low concentrations of Al and Ti
(Table 1). Boström and Peterson (1969) indicated that hydrothermal deposits can
display extremely low $Al/(Al+ Fe+Mn)$ ratios ($< 0.4$), consistent with our results.
Furthermore, the ternary diagrams of $Fe-Mn-(Co + Ni + Cu) \times 10$ of our samples
distinguished a hydrothermal, rather than hydrogenous or diagenetic, origin (Fig. 3a).
Chondrite-normalized REE patterns of the Fe-Si oxyhydroxides showing positive Eu
anomalies and slight LREE enrichment are typical characteristics of high-temperature
hydrothermal fluids (Michard et al., 1983; Craddock et al., 2010). The lack of trace



elements and REEs in the Fe-Si oxyhydroxides of this study indicates that they were
rapidly precipitated from hydrothermal fluids with a small amount of content
scavenged from ambient seawater (German et al., 1990). Mineral composition
analysis also supports this view. In particular, the compositions of 2-line ferrihydrite,
birnessite, pyrite and opal have been identified to be closely related to hydrothermal
activity, as they are consistent with the compositions of other low temperature
hydrothermal deposits (Boyd and Scott, 2001; Hein et al., 2008; Peng et al., 2011).
The precipitation temperatures derived from $^{18}O$ isotope data further support the
mixing and dilution of hydrothermal fluids and seawater.

Fe-Si oxyhydroxides are widespread in modern hydrothermal fields, such as the

East Pacific Rise, Juan de Fuca Ridge, TAG hydrothermal field, Lilliput hydrothermal
field, Wocan hydrothermal field, and Southern Mid-Atlantic ridge (Hekinian et al.,
1993; Mills et al., 1996; Boyd and Scott, 2001; Hrischeva and Scott, 2007; Dekov et
al., 2010; Sun et al., 2012). Hekinian et al. (1993) classified hydrothermal Fe-Si
oxyhydroxides into four types based on their geological setting, morphology,
mineralogy and composition (Fig. 3b). The ternary diagram of Fe-Si-(Co + Ni + Cu +
Zn) x10 indicates that sample 33II-T22 falls within the Fe-rich field of type I (Si/Fe <
0.30, [Co + Ni + Cu + Zn ] > 1000 ppm, Table 1). In contrast, the bulk composition of
sample 20V-T8 is dominated by high Si and low Fe contents, and depleted in trace
elements. Therefore, the 20V-T8 deposit enriched in amorphous opal belongs to type
IV (Si/Fe = 5.4, [Co + Ni + Cu + Zn ] < 1000 ppm, Table 1). Moreover, Fe-Si
oxyhydroxide deposits from samples DIV95-1, DIV95-2, 21V-T7 and 21V-T1 all



show intermediate enrichment in Fe and Si, so plot in the type III field (Si/Fe =
0.50–2.2, Table 1).

Previous studies have put forward different hypotheses for the formation of Fe-Si

deposits in low-temperature hydrothermal environments, including i) the direct
precipitation from hydrothermal fluids (Michard et al., 1984; Alt, 1988; Severmann et
al., 2004), ii) alteration products of sulfides (Iizasa et al., 1998; Chaumba, 2017) or
metalliferous sediments (Fortin et al., 1998; Hrischeva and Scott, 2007), and iii)
biogenic Fe-Si oxyhydroxide (Toner et al., 2009; Devok et al., 2010; Peng et al., 2011;
Bernis et al., 2012; Sun et al., 2012, 2015). Low Fe/Mn and Fe/REE ratios are a
unique feature of samples DIV95-1 and DIV95-2 (Table 2). Some studies have
demonstrated that Fe/Mn and REE/Fe ratios of deposits increased away from a
hydrothermal source (Mitra et al., 1994; German et al., 2002; Edmonds and German,
2004). Therefore, the paragenetic sequences between Fe oxyhydroxides and Mn
oxides of DIV95 may be attributed to the evolution of low-temperature diffuse fluids
in the process of during discharge. This conclusion is supported by geological
evidence from modern and ancient low-temperature hydrothermal fields (Severmann
et al., 2004; Ta et al., 2017). We observed that the $SiO_2$ content (55.32–80.21%) of our
samples was substantially higher than that of Fe-Si oxyhydroxides produced by the
alteration of hydrothermal sulfides (Hekinian et al., 1993; Iizasa et al., 1998). In
addition, sulfur content was measured to be low (0.18–0.39%). These results
suggested that Fe-Si oxyhydroxides in our samples cannot be derived from the
alteration of sulfides (Hekinian et al., 1993). However, observed slight enrichment in



LREE with a pronounced positive Eu anomaly indicated that these deposits were
likely to have formed by direct precipitation from hydrothermal fluids (Michard et al.,
1983). In addition, elevated REE and P content was observed in the purple-red
34II-T22 deposit, which indicated that biogenic Fe-stalks may have played a
significant role in the precipitation of the deposits. Therefore, we propose that
microbes may have contributed to hydrothermal Fe-Si oxyhydroxide formation at the
ultra-slow spreading SWIR.
**5.2 Implications of Sr-Nd-Pb isotope content**
The Sr-Nd-Pb isotopes of low-temperature hydrothermal Fe-Si deposits at the
SWIR show a different sources. In order to map the peculiar isotope signature of our
studied samples, the data have been plotted in Nd/Sr, Sr/Pb and Pb/Pb diagrams.
Sr-Nd-Pb isotopes showed higher $^{87}Sr/^{86}Sr$, $^{207}Pb/^{204}Pb$, $^{208}Pb/^{204}Pb$ and lower
$^{143}Nd/^{144}Nd$ and $^{206}Pb/^{204}Pb$ ratios compared to those of Mid Pacific Ridge (MPR) and
Southeast Indian Ridge (SEIR) basalts (Fig. 7). This result may be closely associated
with the ultra-slow spreading rate and the presence of robust magmatism at the SWIR
(Dupré and Allègre, 1983; Allègre et al. 1984; Meyzen et al., 2005; Yang et al., 2017).
The Sr isotope characteristics of the studied deposits are consistent with direct
precipitation of Fe-Si oxyhydroxides from low-temperature hydrothermal fluids
(Dekov et al., 2010; Yang et al., 2015). This indicates the deposits probably
precipitated mainly from hydrothermal fluids mixed with a subsidiary amount of
ambient seawater (Allègre et al., 1984; Severmann et al., 2004). Furthermore, the
Sr-Nd isotopic plots of the Fe-Si deposits are very similar to the isotopic compositions





observed in the Lilliput and Jan Mayen hydrothermal deposits (Devok et al., 2010;
Johannessen et al., 2016). This probably indicates that the Nd isotope composition of
the Fe-Si deposits was inherited from local parent basalts and seawater. The 34II-T22
and 20V-T8 deposits have particularly pronounced positive εNd values compared with
the other samples (Table 2), which indicates that Nd content reflects the influence of
hydrothermal fluids leaching from substrate rocks. The presence of a positive Eu
anomaly in the Fe-Si deposits further supports this interpretation. We propose that the
Sr and Nd isotope compositions of the Fe-Si deposits at the SWIR might be closely
related to interaction of hydrothermal fluids and seawater.

The distinct Pb isotope compositions in the Fe-Si deposits compared to other

geological settings (Fig. 7) clearly reflects the different isotopic compositions of Pb
sources. The Pb isotope compositions of the studied samples were consistent with
those of basalts from the same part of the SWIR (Yang et al., 2017). This confirms the
role of basalt as a source of Pb in the low-temperature hydrothermal deposits. These
conclusions are supported by the fact that that there was little variation in the Pb
isotope composition of the Fe-Si deposits was observed, due to homogenization by
hydrothermal circulation. Hamelin and Allègre (1985) discussed that Pb isotopic
homogenization can be interpreted in terms of the SWIR being contaminated by a
mantle source. Although melt supply is limited at the ultra-slow spreading SWIR,
plume-ridge interaction may generate significant geochemical anomalies beneath the
SWIR (Breton et al., 2013; Yang et al., 2017). Plume influence at the SWIR is
supported by the presence of thicker crust and hotter mantle between the Indomed and



Gallieni Fracture Zone (Sauter et al., 2009). We infer that the peculiar Pb isotope
composition of Fe-Si deposits might be genetically linked to plume-ridge interactions
at the SWIR.
**5.3 Formation of biogenic Fe-oxyhydroxides at the SWIR**

Microbial activity is a potential starting point for investigating the mechanisms

that have contributed to the formation of hydrothermal Fe-Si deposits (Emerson et al.
2007; Edwards et al. 2011; Johannessen et al., 2016). Previous studies have shown
that Fe(II) oxidation by Fe-oxidizing bacteria results in distinct morphologies of
Fe-oxyhydroxides in hydrothermal deposits, microbial mats, and redox-stratified
water columns (Edwards et al., 2011; Peng et al., 2015; Chan et al., 2016; Chiu et al.
2017). SEM analysis of the Fe-oxyhydroxides in this study indicated that microbes
were widely involved in their precipitation at the SWIR (Fig. 4), with biogenic
Fe-oxyhydroxides in the studied deposits exhibiting various morphologies and sizes.
We observed two different types of rich-Fe biomineralized forms occurring in the
deposits. In particular, abundant sheaths, stalks and filaments enriched in iron
resemble those produced by *Gallionella ferruginea*, *L. ochracea*, and *M. ferrooxydans*
(Edwards et al. 2011; Peng et al., 2015; Chan et al., 2016). Likewise, nanoSIMS ion
mapping of discrete Fe-rich filaments provided further direct evidence for their
formation by Fe oxidizing bacteria (Fig. 8). Furthermore, encrustation of spherical
and rod-like forms clearly indicated that were of biogenic origin (Sun et al., 2015).
The microbes encrusted by Fe-oxyhydroxides may be responsible for biologically
induced mineralization (Ta et al., 2017). We suggest that two types of biomineralized



deposits with distinct forms are produced either directly or indirectly at the SWIR
(Fortin and Langley, 2005; Mikutta et al., 2008; Peng et al., 2015; Chui et al., 2017).

In fact, the reduced iron contributed by hydrothermal systems fuels microbial

activity though the oxidation Fe(II) to Fe(III), which leads to the precipitation of
Fe-bearing minerals (Field et al., 2016; Makita et al., 2016). Some species of microbe
would be viable and active in situ, considering the relative abundance of ferric iron
oxides (oxyhydroxides) $Fe_{OX1}$ and $Fe_{OX2}$, as revealed by iron speciation data (Fig. 6
and Table 4) (Chan et al., 2016; Johannessen et al., 2016). We observed that microbial
Mn(II) oxidization was responsible for the formation of the black layer in sample
DIV95-2, which had a low $Fe_{OX1}$ content. However, as biogenic Fe-oxyhydroxides
were more abundant in samples DIV95-1, 34II-T22, 21V-T7 and 20V-T8, the $Fe_{OX1}$
content also increased. The Mössbauer results further suggested that octahedral Fe(III)
in our samples was the dominant Fe species, which was in good agreement with the
XRD results and iron speciation data (Figs. 5 and S2). No Fe(II) doublets were
detected in any of the samples based on Mössbauer data, indicating the hydrothermal
deposits formed in oxidizing microenvironments. Chan et al. (2016) showed that stalk
and sheath morphologies of Fe mats may reflect the redox conditions, and the
morphologies observed in this study indicate low concentrations of $O_2$ during
formation. Somewhat surprisingly, large amounts of reactive iron minerals such as
2-line-ferrihydrite and lepidocrocite can be identified in all deposits (Figs. 5 and 6).
As a result, bacterial oxidation of dissolved Fe(II) is expected to have produced the
2-line-ferrihydrite and lepidocrocite (Kappler and Newman, 2004; Larese-Casanova



et al., 2010; Chan et al., 2011; Peng et al., 2015). Therefore, as mentioned above,
these findings imply that biomineralization can effectively promote the precipitation
of iron bearing minerals in modern and ancient hydrothermal fields.
**5.4 Formation of biogenic silica**

Well-preserved morphologies of biomineralized silica were common in all the

Fe-Si deposits at the SWIR (Fig. 6), representing a link between oceanic crust and life
(Zierenberg et al., 2000; Conley et al., 2017). The dissolved silica is thought to be
primarily derived from hydrothermal fluids, based on the expected composition of
hydrothermal fluids. Silica is not an essential nutrient for microbes found at
hydrothermal sites (Baross and Hoffman, 1985; Martin et al., 2008). Based on mineral
phase relationships and temperatures of precipitation deduced from stable isotopes of
O, we infer that the precipitation of silica is probably driven by microbial activity.
Morphologically similar sheaths, stalks, filaments, spheroidal and rod-like forms
within the deposits may be regarded as biosignatures that can survive in
low-temperature environments. The abundant structurally coordinated silica may
allow the preservation of microbial primary features. This interpretation is in
accordance with those of encased microbes observed in banded iron formations and
ancient jaspers (Chi Fru et al., 2013; Grenne and Slack, 2003). Of particular note is
the presence of abundant poorly reactive sheet silicate iron ($Fe_{PRS}$) in our samples
(Table 4). It is likely that biogenic Fe-Si oxyhydroxides have been transformed into
ordered poorly reactive sheet silicates, such as nontronite (Ueshima and Tazaki, 2001;
Dekov et al., 2007; Sun et al., 2011). Previous studies have put forward two



hypotheses for the precipitation of biogenic silica. Firstly, microbes may serve as
reactive geochemical surfaces where Si is directly adsorbed and precipitated (Juniper
and Fouquet, 1988; Ueshima and Tazaki, 2001; Jones et al., 2004; Peng and Jones,
2012). We observed many silicified spherical and rod-like microbes preserved in the
DIV-95-1 and 21V-T1 deposits (Figs. 6a and 6d). Biomineralization experiments
performed using a variety of marine microorganisms have demonstrated that
unsheathed bacteria can become encrusted in silica (Orange et al., 2009; Li et al.,
2013). The second hypothesis suggests interactions between Fe oxyhydroxide and
silica occur as a result of microbial activity (Dupraz and Visscher, 2005; Peng et al.,
2011; Sun et al., 2015). For instance, we found preformed silica colloids, which
measured few tens of nanometers in diameter, attached to the surface of Fe oxidizing
sheaths. Upon aging, the structurally coordinated Fe(III) becomes partially replaced
by amorphous Si, and is transformed into ordered Fe-Si oxyhydroxides (Fein et al.,
2002; Pokrovski et al., 2003; Devok et al., 2010). This interpretation is in accordance
with the existence of filamentous microfossils found in submarine hydrothermal vent
precipitates more than 3,770 million years ago (Dodd et al., 2017). We propose that
the geochemical constituents of mineralized microbes imply that dissolved silica and
ferric iron were original reactants in the low-temperature hydrothermal systems of the
SWIR. Given the morphological, mineralogical and geochemical characteristics of the
deposits, the ultra-slow spreading SWIR might be regarded as a potential region for
the origin and evolution of life.





**6 Conclusions**


Fe-Si deposits collected from the ultra-slow spreading SWIR showed evidence of
low-temperature hydrothermal origin. The deposits were mainly composed of 2-line
ferrihydrite, pyrite, natrojarosite and amorphous opal, characterized by both negative
Ce and positive Eu anomalies, along with a slight enrichment in LREE. The Sr-Nd-Pb
isotopic compositions of the Fe-Si deposits were partially inherited from a mantle
source mixed with seawater by low-temperature hydrothermal circulation. Two
different types of biomineralized forms were preserved in Fe-Si deposits. These
biogenic Fe-Si oxyhydroxides clearly showed that microbial activity played a
significant role in the formation of the hydrothermal deposits, either directly or
indirectly, due to biologically-induced mineralization. Mössbauer spectra and iron
speciation data provided further insight into the iron-bearing phases in these deposits.
These findings supported the hypothesis that microbial activity was the principal
deposition mechanism of Fe-Si oxyhydroxides in modern and ancient seafloor
hydrothermal systems. Such studies shed light on the possibility that the origin and
evolution of life in an environment similar to the ultra-slow spreading SWIR.

*Data availability*. The data of the different experiments are freely available upon
request from the corresponding author. Data sets supporting the results are also
archived in an open-access database: https://doi.org/10.6084/m9.figshare.9521117.v1.

*Author contributions*. This work was conceived and supervised by ZW and XP, ZL



and KT performed the measurements and data evaluation. KT wrote the paper with
contributions from all coauthors.

*Competing interests*. The authors declare that they have no conflict of interest.

*Acknowledgments*. Special thanks go to all of the participants of the cruise of the R/V
DaYang YiHao and XYH09 conducted by the China Ocean Mineral Resource R&D
Association (COMRA). This work is directly supported by the National Nature
Science Foundation of China (Grant No. 2016YFA0601100) and the National Natural
Science Foundation of China ( Grant No.41176065). NanoSIMS analyses were
performed at the Institute of Geology and Geophysics, Chinese Academy of Sciences.
The Sr, Nd, and Pb isotopic compositions were performed in the Laboratory for
Radiogenic Isotope Geochemistry at the University of Science and Technology of
China.

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

961

962

963





**Figure captions**

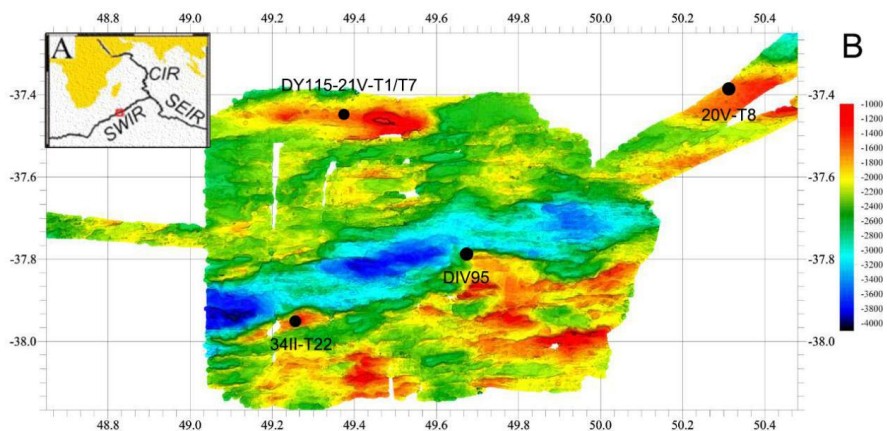

Figure 1. Regional bathymetric map and location of the sampling site at the SWIR.

Black dots represent sample locations in this study.

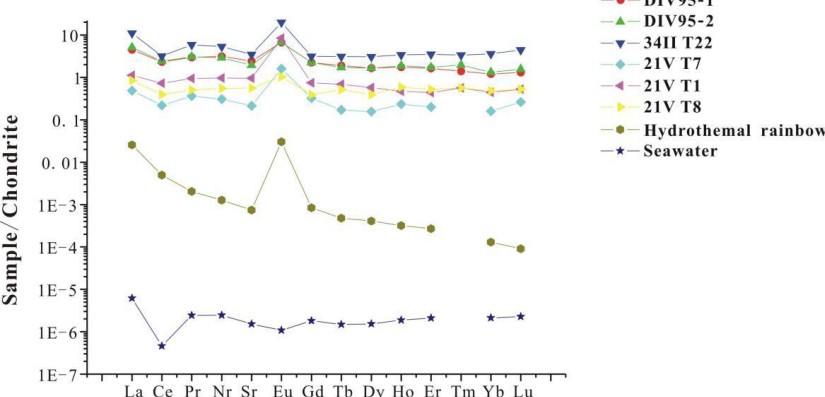

Figure 2. Chondrite-normalized REE distribution patterns of the hydrothermal Fe-Si

deposits in this study.





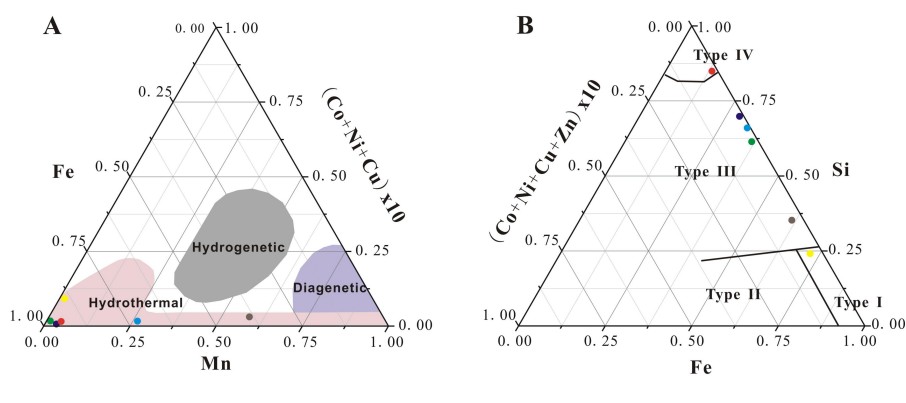

Figure 3. (a) Ternary diagram for Mn-Fe-(Co + Ni + Cu) ×10 of Fe-Si deposits. The
hydrothermal, diagenetic and hydrogenous fields were classified by Hein et al. (1994).
The Fe-Si deposits of this study are inferred to be of hydrothermal origin. (b)
Fe-Si-(Co + Ni + Cu + Zn) ×10 ternary diagram showing the various types of Fe-Si
deposits from Hekinian et al. (1993).

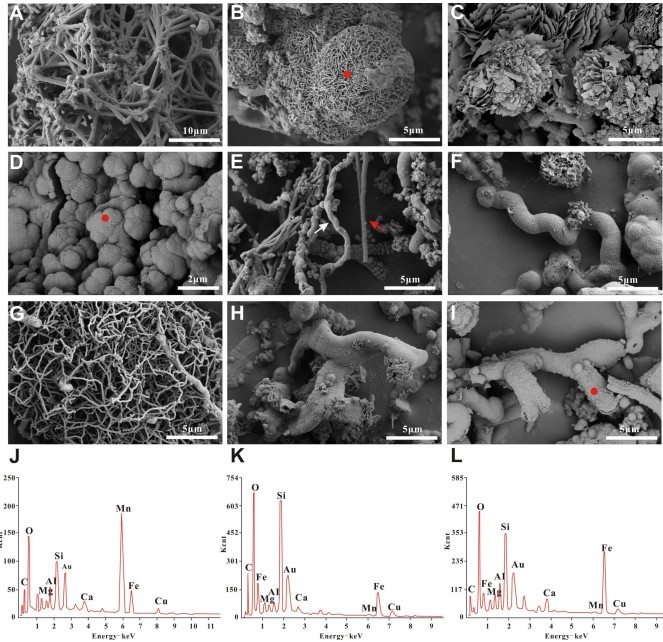

Figure 4. SEM images showing different styles of biogenic mineral structures in
different Fe-Si deposits. (a) A network-like structure composed of rod-like

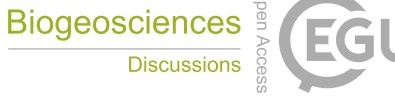

mineralized forms observed in the orange-yellowish sample DIV95-1. (b) Spheroidal
Mn-oxide surface showing honeycomb microtextures in the black sample DIV95-2. (c)
Typically rosette spherical structures observed in the DIV95-2 deposits. (d) Spherical
morphologies encrusted by iron and silicon in the yellowish sample 21V-T1. (e)
Ribbon-like helical filaments (white arrow) and hollow tubes (red arrow) found in the
purple-red sample 34II-T22. (f) Twisted stalk observed in the 34II-T22 deposit. (g) A
network-like structure composed of threadlet filaments observed in the brown sample
21V-T7. (h) Branched sheaths observed in the brown sample 20V-T8. (i) Spheroids
scattered on the surface of a branched sheath found in the brown sample 20V-T8. (j)
EDS from the area defined by the red dot in panel b. (k) EDS from the area defined by
the red dot in panel d. (l) EDS from the area defined by the red dot in panel i.

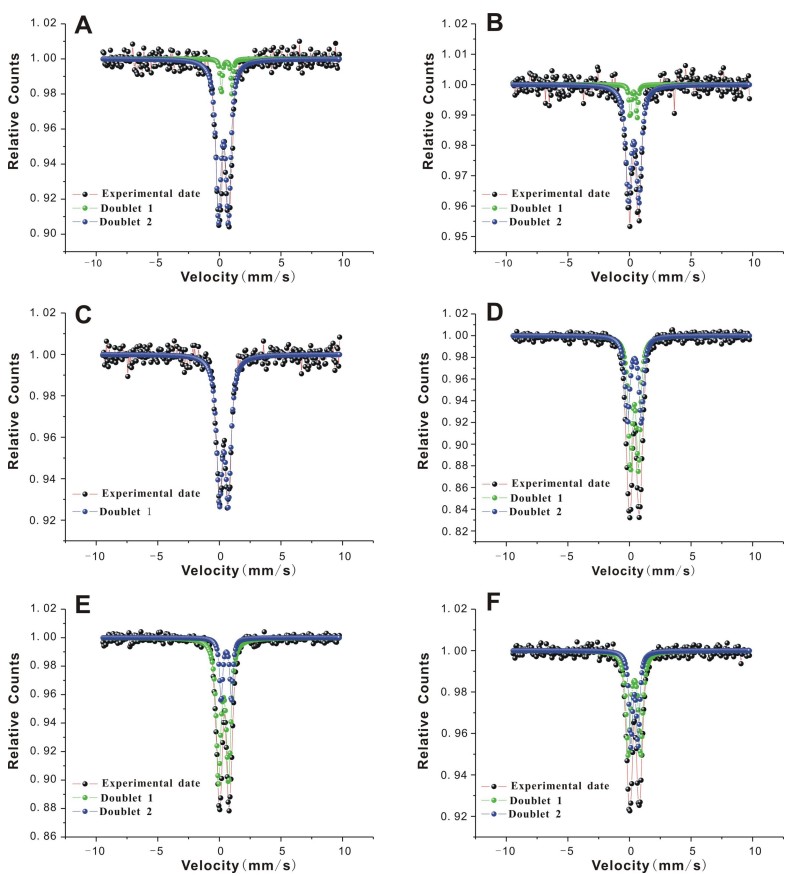

Figure 5. $^{57}$Fe Mössbauer spectra at room temperature (300 K), and fitting results of





990 Fe-Si deposits from the SWIR. (a) DIV95-1, (b) DIV95-2, (c) 34II-T22, (d) 21V-T1,

991 (e) 21V-T7, (f) 20V-T8.

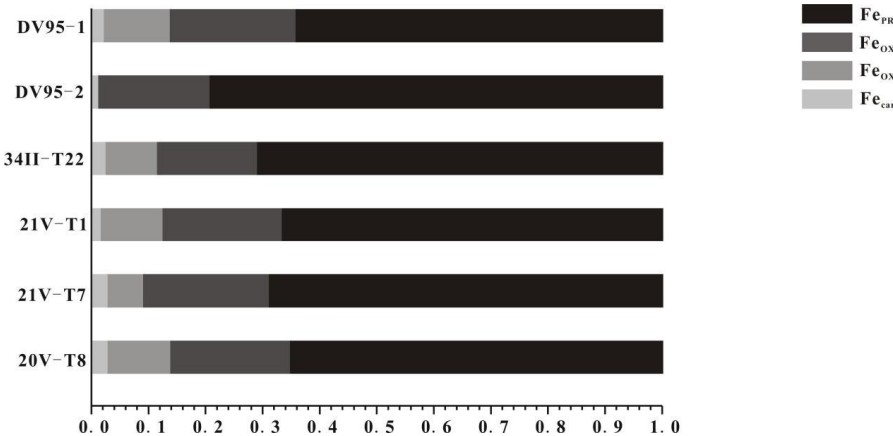

992 Figure 6. Sequential extraction of iron minerals in the studied Fe-Si deposits.

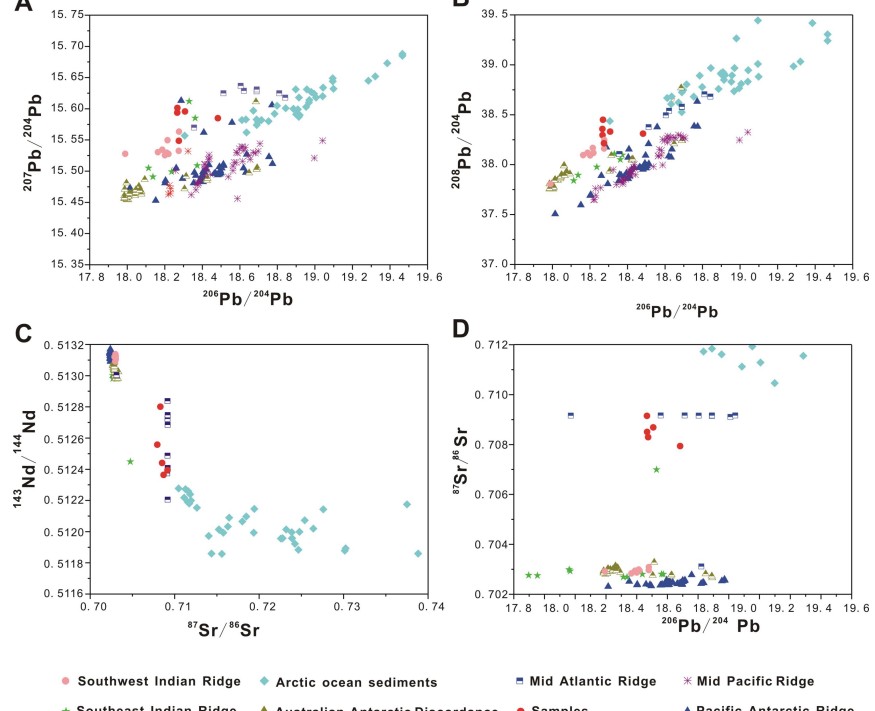

993 Figure 7. Comparison of $^{208}$Pb/$^{204}$Pb versus $^{206}$Pb/$^{204}$Pb (a), $^{207}$Pb/$^{204}$Pb versus

994 $^{206}$Pb/$^{204}$Pb (b), $^{87}$Sr/$^{86}$Sr versus $^{143}$Nd/$^{144}$Nd (c), and $^{87}$Sr/$^{86}$Sr versus $^{206}$Pb/$^{204}$Pb from





the studied Fe-Si deposits, compared against Pacific Ridge basalts (Vlastèlic et al.,
1999), Arctic Ocean sediments (Maccali et al., 2018), Southeast Indian Ridge basalts
(Hamelin and Allègre, 1985), Australian-Antarctic Discordance basalts (Kempton et
al., 2002), Pacific hydrothermal sulfides (Fouquet and Marcoux, 1995), Atlantic
Ridge hydrothermal deposits (Dekov et al., 2010), and Southwest Indian Ridge basalts
(Yang et al., 2017).

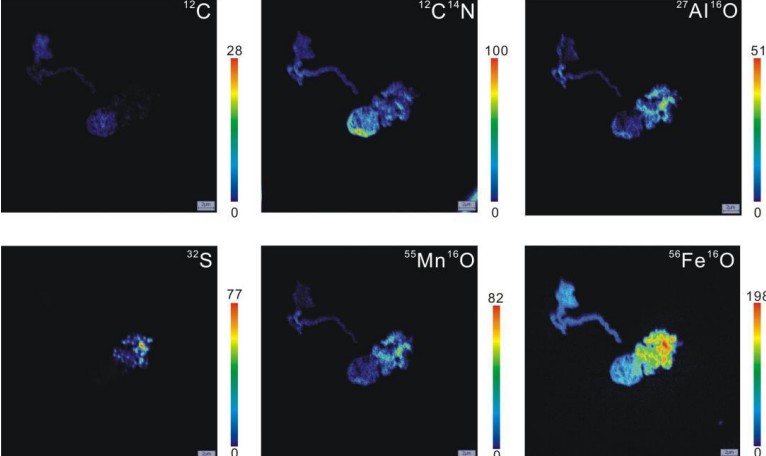

Figure 8. NanoSIMS ionic images of $^{12}C^-$, $^{12}C^{14}N^-$, $^{32}S^-$, $^{27}Al^{16}O^-$, $^{55}Mn^{16}O^-$, and
$^{56}Fe^{16}O_2^-$ from a twisted stalk. Ion intensity variations are shown by calibration bars.
The scale bar is 2 μm for each panel.





**Table 1.** Chemical composition (XRF and ICP-MS) of hydrothermal Fe-Si deposits at the SWIR.

| Sample# | Al₂O₃ (wt.%) | CaO | Fe₂O₃ | K₂O | MgO | MnO | Na₂O | P₂O₅ | TiO₂ | SiO₂ | CuO | SO₃ | Cr₂O₃ | Co₃O₃ | ZnO | Al/(Al+Fe+Mn) | Fe/Mn | TOTAL |
|---|---|---|---|---|---|---|---|---|---|---|---|---|---|---|---|---|---|---|
| DIV95-1 | 0.089 | 1.013 | 23.257 | 0.501 | 0.641 | 7.356 | 4.620 | 0.453 | 0.005 | 55.316 | 0.083 | 0.377 | 0.069 | 0.049 | 0.025 | 0.002 | 2.78 | 95.63 |
| DIV95-2 | 0.096 | 1.704 | 32.964 | 0.581 | 0.851 | 25.467 | 4.777 | 0.264 | 0.007 | 27.220 | 0.159 | 0.180 | 0.063 | 0.073 | - | 0.001 | 1.41 | 99.20 |
| 34II-T22 | 0.031 | 0.696 | 64.733 | 0.06 | 1.396 | 0.124 | 2.101 | 0.898 | 0.004 | 27.076 | 0.294 | 0.295 | 0.006 | 0.018 | 0.177 | 0.001 | 460.17 | 97.90 |
| 21V-T7 | 0.053 | 0.382 | 21.695 | 0.278 | 0.591 | 0.741 | 3.865 | 0.149 | 0.004 | 68.356 | - | 0.396 | - | 0.024 | 0.05 | 0.002 | 25.81 | 98.85 |
| 21V-T1 | 0.072 | 0.506 | 23.216 | 0.294 | 0.664 | 0.516 | 3.364 | 0.356 | 0.004 | 68.106 | - | 0.307 | - | 0.061 | 0.040 | 0.002 | 39.65 | 97.73 |
| 20V-T8 | 0.059 | 0.387 | 10.562 | 0.182 | 0.237 | 0.124 | 1.892 | 0.335 | 0.004 | 85.205 | - | 0.184 | - | 0.026 | 0.0235 | 0.002 | 75.08 | 99.61 |

| Sample# | Li (ppm) | Be | Sc | V | Cr | Co | Ni | Cu | Zn | Rb | Sr | Y | Zr | Nb | Mo | In | Cs | Ba | Tl |
|---|---|---|---|---|---|---|---|---|---|---|---|---|---|---|---|---|---|---|---|
| DIV95-1 | 55.325 | 0.230 | 0.669 | 87.225 | 4.104 | 13.210 | 27.647 | 313.016 | 155.360 | 7.618 | 230.843 | 3.994 | 2.067 | 0.110 | 115.621 | 0.030 | 0.399 | 254.107 | 2.636 |
| DIV95-2 | 263.918 | 0.020 | 4.789 | 158.651 | 4.928 | 42.197 | 54.383 | 1095.661 | 253.322 | 8.297 | 343.394 | 4.159 | 4.079 | 0.240 | 420.370 | 0.070 | 0.370 | 682.089 | 4.259 |
| 34II-T22 | 1.747 | 0.220 | 0.988 | 675.981 | 16.069 | 232.846 | 23.085 | 1960.176 | 1478.588 | 1.757 | 296.921 | 10.290 | 10.939 | 0.409 | 168.771 | 0.240 | 0.269 | 46.889 | 2.575 |
| 21V-T7 | 1.454 | 0.538 | 2.609 | 71.919 | 3.355 | 0.767 | 2.589 | 10.514 | 79.814 | 2.858 | 221.541 | 0.438 | 1.195 | 0.090 | 82.373 | 0.010 | 0.279 | 943.514 | 0.149 |
| 21V-T1 | 6.082 | 0.739 | 0.210 | 39.647 | 1.618 | 1.059 | 4.154 | 40.396 | 126.830 | 1.348 | 253.959 | 1.128 | 2.127 | 4.983 | 42.773 | 0.010 | 0.040 | 926.755 | 0.499 |
| 20V-T8 | 18.321 | 0.080 | 0.360 | 39.871 | 1.010 | 1.279 | 3.848 | 69.487 | 44.309 | 1.769 | 141.332 | 1.129 | 1.059 | 0.740 | 22.189 | 0.000 | 0.020 | 420.797 | 0.750 |






| Sample# | Pb (ppm) | Bi | Th | U | Hf | Ta | La | Ce | Pr | Nd | Sm | Eu | Gd | Tb | Dy | Ho | Er | Tm | Yb |
|---|---|---|---|---|---|---|---|---|---|---|---|---|---|---|---|---|---|---|---|
| DIV95-1 | 2.257 | 0.070 | 0.030 | 12.601 | 0.040 | 0.170 | 1.657 | 2.227 | 0.409 | 2.237 | 0.559 | 0.579 | 0.689 | 0.110 | 0.629 | 0.150 | 0.409 | 0.050 | 0.300 |
| DIV95-2 | 3.609 | 0.080 | 0.050 | 11.446 | 0.080 | 0.380 | 1.869 | 2.319 | 0.420 | 2.059 | 0.450 | 0.590 | 0.690 | 0.100 | 0.630 | 0.160 | 0.430 | 0.070 | 0.330 |
| 34II-T22 | 54.065 | 0.180 | 0.100 | 43.854 | 0.130 | 0.289 | 4.082 | 3.054 | 0.808 | 3.813 | 0.808 | 1.727 | 0.968 | 0.180 | 1.168 | 0.289 | 0.878 | 0.120 | 0.898 |
| 21V-T7 | 1.932 | 0.020 | 0.010 | 6.681 | 0.030 | 0.199 | 0.179 | 0.209 | 0.050 | 0.219 | 0.050 | 0.139 | 0.100 | 0.010 | 0.060 | 0.020 | 0.050 | 0.000 | 0.040 |
| 21V-T1 | 1.818 | 0.080 | 0.040 | 6.611 | 0.030 | 3.126 | 0.409 | 0.699 | 0.130 | 0.689 | 0.220 | 0.739 | 0.230 | 0.040 | 0.220 | 0.040 | 0.110 | 0.020 | 0.110 |
| 20V-T8 | 4.558 | 0.040 | 0.010 | 0.510 | 0.020 | 0.850 | 0.310 | 0.380 | 0.070 | 0.390 | 0.130 | 0.090 | 0.120 | 0.030 | 0.150 | 0.050 | 0.130 | 0.020 | 0.120 |

| Sample# | Lu (ppm) | ΣREE | LREE | HREE | Eu/Eu* | Ce/Ce* | $La_N/Yb_N$ | $La_N/Sm_N$ | Fe/ΣREE |
|---|---|---|---|---|---|---|---|---|---|
| DIV95-1 | 0.050 | 10.054 | 7.668 | 2.386 | 2.850 | 0.644 | 3.74 | 1.86 | 1.57 |
| DIV95-2 | 0.060 | 10.177 | 7.708 | 2.469 | 3.230 | 0.616 | 3.83 | 2.61 | 2.20 |
| 34II-T22 | 0.170 | 18.963 | 14.292 | 4.671 | 5.957 | 0.388 | 3.07 | 3.17 | 2.32 |
| 21V-T7 | 0.010 | 1.135 | 0.846 | 0.289 | 5.935 | 0.534 | 3.04 | 2.26 | 12.99 |
| 21V-T1 | 0.020 | 3.675 | 2.886 | 0.789 | 9.979 | 0.738 | 2.81 | 1.17 | 4.29 |
| 20V-T8 | 0.020 | 2.009 | 1.369 | 0.640 | 2.165 | 0.607 | 1.74 | 1.50 | 3.91 |

$Ce/Ce^* = 2Ce_N/(La_N + Pr_N)$; $Eu/Eu^* = 2Eu_N/(Sm_N + Gd_N)$






**Table 2.** Pb, Sr, Nd, and O isotopic data for studied samples and deduced temperature.

| Sample[#] | $^{87}Sr/^{86}Sr$ | (2σ) | $^{143}Nd/^{144}Nd$ | (2σ) | $^{206}Pb/^{204}Pb$ | (2σ) | $^{207}Pb/^{204}Pb$ | (2σ) | $^{208}Pb/^{204}Pb$ | (2σ) | $^{147}Sm/^{144}Nd$ | εNd | $δ^{18}O$ (‰ SMOW) | Deduced temperture (°C) |
|---|---|---|---|---|---|---|---|---|---|---|---|---|---|---|
| DIV95-1 | 0.708509 | 0.000014 | 0.512441 | 0.000011 | 18.2674 | 0.0029 | 15.6013 | 0.0036 | 38.2964 | 0.0119 | 0.1416 | -3.8 | 20.908 | 43.23 |
| DIV95-2 | 0.708686 | 0.000013 | 0.512364 | 0.000011 | 18.3076 | 0.0027 | 15.5957 | 0.0029 | 38.3313 | 0.0096 | 0.1298 | -5.3 | / | / |
| 34I1-T22 | 0.70915 | 0.000012 | 0.512801 | 0.000015 | 18.2664 | 0.002 | 15.5939 | 0.002 | 38.3574 | 0.0061 | 0.1180 | 3.2 | 35.87 | 114.66 |
| 21V-T1 | 0.708519 | 0.000015 | 0.512332 | 0.000015 | 18.2694 | 0.0016 | 15.6404 | 0.0036 | 38.4493 | 0.0062 | 0.1645 | -6 | 17.352 | 31.17 |
| 21V-T7 | 0.707937 | 0.000013 | 0.512578 | 0.000031 | 18.4829 | 0.0026 | 15.5851 | 0.0028 | 38.3103 | 0.0092 | 0.1924 | -1.2 | 16.564 | 28.76 |
| 20V-T8 | 0.708297 | 0.000012 | 0.512801 | 0.000015 | 18.2749 | 0.0009 | 15.5488 | 0.0011 | 38.2145 | 0.0034 | 0.1721 | 3.2 | 20.525 | 41.83 |





Table 3. Mössbauer parameters (room temperature) of hydrothermal Fe-Si deposits at
the SWIR.

| Sample# | IS | QS | LW | A (%) | Assignment | Mineralogy |
|---|---|---|---|---|---|---|
| DIV95-1 | 0.59 | 0.87 | 0.19 | 8.50 | $^{VI}Fe^{3+}$ | **Octahedral ferrihydrite** |
| | 0.33 | 0.79 | 0.49 | 91.50 | $^{VI}Fe^{3+}$ | **2-line-ferrihydrite** |
| DIV95-2 | 0.34 | 0.55 | 0.19 | 10.40 | $^{IV}Fe^{3+}$ | **Lepidocrocite** |
| | 0.34 | 0.85 | 0.50 | 89.60 | $^{VI}Fe^{3+}$ | **2-line-ferrihydrite** |
| 34II-T22 | 0.36 | 0.72 | 0.53 | 100.00 | $^{VI}Fe^{3+}$ | **2-line-ferrihydrite** |
| 21V-T1 | 0.40 | 0.65 | 0.40 | 57.40 | $^{VI}Fe^{3+}$ | **Goethite** |
| | 0.40 | 1.11 | 0.44 | 42.60 | $^{VI}Fe^{3+}$ | **Octahedral ferrihydrite** |
| 21V-T7 | 0.34 | 0.71 | 0.43 | 75.70 | $^{VI}Fe^{3+}$ | **2-line-ferrihydrite** |
| | 0.56 | 0.84 | 0.3 | 24.30 | $^{VI}Fe^{3+}$ | **Octahedral Fe(III)** |
| 20V-T8 | 0.40 | 1.01 | 0.42 | 58.70 | $^{VI}Fe^{3+}$ | **Octahedral ferrihydrite** |
| | 0.40 | 0.58 | 0.33 | 41.30 | $^{VI}Fe^{3+}$ | **Lepidocrocite** |

A=relative spectral area, IS=isomer shift, QS=quadrupole splitting, LW=line-width.





Table 4. Relative compositions of Fe-bearing minerals of hydrothermal Fe-Si deposits
at the SWIR.

| Sample[#] | $Fe_{carb}$ (μmol/g) | $Fe_{ox1}$ (μmol/g) | $Fe_{ox2}$ (μmol/g) | $Fe_{PRS}$ (μmol/g) |
|---|---|---|---|---|
| DIV95-1 | 19.83 | 116.43 | 221.07 | 647.46 |
| DIV95-2 | 11.38 | 0.40 | 223.21 | 910.60 |
| 34II-T22 | 28.70 | 111.71 | 218.13 | 885.64 |
| 21V-T1 | 15.60 | 117.16 | 225.90 | 723.42 |
| 21V-T7 | 27.11 | 63.95 | 226.51 | 710.03 |
| 20V-T8 | 28.27 | 118.48 | 224.89 | 703.17 |







