# Peer review of "Southwest Indian Ridge"

_Biogeosciences, 2019_

## Referee Comment (RC1) · Anonymous Referee #1 · 17 Jan 2020

The manuscript by Kaiwen Ta et al. presents a rather exhaustive panoply of analytical techniques performed on samples from six exhalative Fe-Si deposits collected during a cruise in the SW Indian Ridge. The authors conclude that their analysis show that these deposits are of low temperature, mainly made of Fe-Si and that there is a strong biological influence in their formation. They also present Sr-Nd-Pb isotope that seem to support their conclusions. The topic can be of major interest for the readers of Biogeosciences. However, my opinion is that the manuscript needs major and complete rewritting before being considered for publication – is too repetitive, phrases are vague and is not clearly shown what these techniques really add to the state of the art. Writing is extremely repetitive and the text should be checked by an English-speaking special-

ist. Also, I have the feeling that the authors have used different exciting and novel techniques but without having a clear focus in what they try to show. I would suggest to careful evaluate if the use of these techniques adds something to the interpretation of these rocks that could well easily done with some basic geology and a conventional petrographic-chemical analysis. Finally, I have serious doubts that the authors are able to prove that these structures found in the iron oxydehydroxyde deposits and silica precipitates represent fossilized microbes despite your are within the life thermal window. By aware that inorganic silica growth, just an example, can mask organic textures (see, for example, Garcia Ruiz et al 2017, Science). For being sure that these structures represent past organic activity you must show TEM images and/or some stable isotopes indicative of biogenic-promoted redox equilibria. If in active sites, you should try some geomicrobiological studies. Finally, all the radiogenic isotope geochemistry needs some reinterpretation. Pb isotopes are not just indicative of major hydrothermal activity -that is saying nothing in terms of radiogenic isotope geochemistry. The statement that Sr-Nd isotopes "were closely related to interaction between hydrothermal fluids and seawater" is also ambiguous. Obviously, there must a significant part of the Sr inherited from seawater but the hydrothermal fluids must transport some also. Nd is unlikely to be derived from seawater and perhaps the Nd isotope signature should be controlled by the hydrothermal fluid – mixing diagrams are fundamental for this discussion. But a key unresolved question is where the deep fluids come from? Probably they are equilibrated with oceanic crust but this needs to be discussed. Please check ambiguous phrases such as "appropriate solvent" or be aware of the analytical error when quoting stable isotopes – you cannot go to the second decimal. Also, you cannot go to the second!! decimal when calculating isotope temperatures. Errors here are usually above $\pm 20°C$. You have to explain how this was calculated. The same holds true for Pb isotopes... 4th decimal!! Please, have all these data checked by an specialist in isotope geochemistry. You say that positive eNd values are indicative of a mantle derivation – that is ok but you can say a lot more with your data. And what about the negative values? Your reservoir looks really heterogeneous and this ample

range of eNd values need to be discussed. The same for Sr isotopes. Your range of data is extremely variable, is not a "slight variation" going from 0.7079 to 0.7091 and you need to explain this – not done in the manuscript. Also, you are talking about oxidized systems and you quote pyrite by XRD. How abundant is the pyrite? Where it is located? Is it the primary mineral that has been extensively oxidized? Or pyrite is just a local precipitate in a more anoxic setting? The low S contents today do not prove that these rocks were originally precipitated as sulphide rocks and later being oxidized. You must have stronger arguments. These questions need to be solved by just careful observations before performing a batch of uncontrolled analytical techniques. Also, you must try to interpret all your results, even if contradictory, not just some of them. Unless you unambiguously prove direct or indirectly, that the structures are microbial the major conclusions of the paper should be considered just an attractive but plausible hypothesis. Don't go too far into speculative conclusions before being sure of that. The discussion needs to be completely rewritten but probably the aforementioned aspects need to be solved before getting into a thoughtful review.

―――――――――――――――――――――

---

## Referee Comment (RC2) · Anonymous Referee #2 · 12 Feb 2020

The paper "Formation and origin of Fe-Si oxyhydroxide deposits at the ultra-slow spreading Southwest Indian Ridge" by Ta et al. is a geochemical study of six Fe and Si rich samples from the SWIR, collected by various means between 2008 and 2015. Although the data appears well collected and analytical work is extensive and appears sound, the context of the paper, including the introduction, interpretation of results and discussion is jumbled, and there needs to be more discussion reconciling the different analytical results. For example, the low sulfur content (line 436) is inconsistent with the presence of pyrite as a major mineral (line 286); also none of these major minerals (line 286) represent phyllosilicates, stated as the most abundant Fe pool obtained by the leach procedure (line 340). Also, the Mössbauer shows exclusively Fe(III), however

[Figure]

XRD reveals pyrite, Fe(II). The authors claim goethite and hematite are mineral phases (line 337-338), but do not see these via XRD, The paper does not address the possible reasons for these inconsistencies, and must.

The discussion is also internally inconsistent. For example, regarding the role of microbial activity in forming the Fe-Si minerals line 582 suggests "these findings support the hypothesis that microbial activity was the principal deposition mechanism of Fe-Si oxyhydroxides in modern and ancient seafloor hydrothermal systems" while line 494 notes microbes "were widely involved" and line 442 suggests that they "may have played a role." The conclusions (line 584-585) and line 567-569 attempt to tie in the "origin and evolution of life" which isn't discussed in the rest of the paper and seems to be a non sequitur to the rest of the manuscript. Additionally, the paper should be proofread for grammatical issues and other issues, for instance there is no "Mid Pacific Ridge", the scale bars in Figure 8 are really not visible.

---

## Author Comment (AC1) · 3 Mar 2020

Response to Reviewer

All of us would like to thank the editor and reviewer again for your considerations about this manuscript and constructive comments. We have revised it very carefully. Responses are provided in a point-by-point fashion.
RC1: The manuscript by Kaiwen Ta et al. presents a rather exhaustive panoply of analytical techniques performed on samples from six exhalative Fe-Si deposits collected during a cruise in the SW Indian Ridge. The authors conclude that their analysis show that these deposits are of low temperature, mainly made of Fe-Si and that there is a strong biological influence in their formation. They also present Sr-Nd-Pb isotope that seem to support their conclusions. The topic can be of major interest for the readers of Biogeosciences. However, my opinion is that the manuscript needs major and complete rewritting before being considered for publication – is too repetitive, phrases are vague and is not clearly shown what these techniques really add to the state of the art. Writing is extremely repetitive and the text should be checked by an English-speaking specialist.

Answer: Thanks for the reviewer's constructive comments. In the revised version, we have rewritten the manuscript to enhance readability based on your suggestion. We have made considerable efforts to improve the grammar of manuscript.

RC1: Also, I have the feeling that the authors have used different exciting and novel techniques but without having a clear focus in what they try to show. I would suggest to careful evaluate if the use of these techniques adds something to the interpretation of these rocks that could well easily done with some basic geology and a conventional petrographic-chemical analysis.

Answer: Thanks for the reviewer's constructive comments. We have added more discussions on some basic geology and a conventional petrographic-chemical analysis to the interpretation of these rocks in the revised paper.

RC1: Finally, I have serious doubts that the authors are able to prove that these structures found in the iron oxydehydroxyde deposits and silica precipitates represent fossilized microbes despite your are within the life thermal window. By aware that inorganic silica growth, just an example, can mask organic textures (see, for example, Garcia Ruiz et al 2017, Science). For being sure that these structures represent past

organic activity you must show TEM images and/or some stable isotopes indicative of biogenic-promoted redox equilibria. If in active sites, you should try some geomicrobiological studies.

Answer: Thanks for providing valuable information on the biogenic structures distinguishing from inorganic silica growth. We have added TEM images and EDS date to discuss these Fe oxyhydroxide stalks potentially represented past organic activity (Fig. S3). We have also tried some geomicrobiological studies to culture Fe-oxidizing bacteria according to the reviewer's suggestion. We observed that Fe-stalks were produced by Fe-oxidizing bacteria similar to morphologies of Fe-Si oxyhydroxides in studied deposits (Fig. S4). We suggested that these structures found in the iron oxydehydroxyde deposits and silica precipitates represented the microbes encrusted by Fe-Si oxydehydroxydes.

RC1: Finally, all the radiogenic isotope geochemistry needs some reinterpretation. Pb isotopes are not just indicative of major hydrothermal activity -that is saying nothing in terms of radiogenic isotope geochemistry.

Answer: Thanks for this valuable comment. We have rewritten the manuscript on all the radiogenic isotope geochemistry in the revised paper.

RC1: The statement that Sr-Nd isotopes "were closely related to interaction between hydrothermal fluids and seawater" is also ambiguous.

Answer: Thanks for your consideration. We have revised them in the manuscript as following: Sr and Nd isotope compositions of Fe-Si oxyhydroxide deposits at the SWIR probably reflected a combined signature of the hydrothermal fluid and seawater.

RC1: Obviously, there must a significant part of the Sr inherited from seawater but the hydrothermal fluids must transport some also.

Answer: We agree with the reviewer that there is a significant part of the Sr in the Fe-Si oxyhydroxide deposits at the SWIR inherited from seawater, but the hydrothermal fluids

must also transport some.

RC1: Nd is unlikely to be derived from seawater and perhaps the Nd isotope signature should be controlled by the hydrothermal fluid – mixing diagrams are fundamental for this discussion. But a key unresolved question is where the deep fluids come from? Probably they are equilibrated with oceanic crust but this needs to be discussed.

Answer: Thanks for the reviewer's constructive comments. We have added hydrothermal fluid – mixing diagrams to discuss on Nd isotope signature (Fig. 8). We have also added more discussions that the deep fluids probably acted to equilibrate the hydrothermal fluid with oceanic crust, according to the reviewer's suggestions in the revised paper.

RC1: Please check ambiguous phrases such as "appropriate solvent" or be aware of the analytical error when quoting stable isotopes – you cannot go to the second decimal.

Answer: Thank you for pointing them out. We have corrected them in the revised paper as following: The different solvents would subsequently extract the following phases: 1 M Na-acetate, 1 M hydroxylamine-HCl, 0.28 M Na-dithionite and 12 M HCl, for a defined period of time (Table S2).

RC1: Also, you cannot go to the second!! decimal when calculating isotope temperatures. Errors here are usually above $\pm20$ âŮę C. You have to explain how this was calculated. The same holds true for Pb isotopes... 4th decimal!! Please, have all these data checked by an specialist in isotope geochemistry.

Answer: Sorry for these mistakes. We have corrected them in the revised paper (Table 2).

RC1: You say that positive eNd values are indicative of a mantle derivation – that is ok but you can say a lot more with your data. And what about the negative values? Your reservoir looks really heterogeneous and this ample range of eNd values need to be

discussed.

Answer: Thanks for this valuable comment. We have added more discussions on this ample range of eNd values in the revised paper as following:

The mixing between hydrothermal fluids and seawater was likely reflected by Nd isotopic data of the deposits. In order to better understand the mixing relationship between hydrothermal fluids and seawater, ÉŻNd versus 87Sr/86Sr correlation for Fe-Si oxyhydroxide deposits was compared to hydrothermal fluid and seawater in Figure 8. The Nd-Sr isotope mixing model indicated that samples 20V-T8 and 34II-T22 had higher ÉŻNd values (3.2 and 5.1) than modern Indian Ocean seawater values (−8.0) (Pomiès et al., 2002) and were close to the values of hydrothermal fluid (7.1) (Amini et al., 2008). We inferred that Nd in two samples was mainly derived from hydrothermal fluid, which was produced by seawater leaching basalt at elevated temperatures (Fig. 8). The presence of a positive Eu anomaly in the Fe-Si oxyhydroxide deposits further supported this interpretation. On the other hand, negative ÉŻNd values indicated incorporation of abundant seawater Nd into the Fe-Si oxyhydroxide deposits. This result may attribute to an extremely low extent of seawater-basalt interaction during the hydrothermal circulation. Alternatively, negative ÉŻNd values of DIV95-1, DIV95-2, 21V-T1 and 21V-T7 samples possibly reflected the interaction between Fe-Si oxyhydroxides and seawater (Clauer et al., 1984). Baker et al. (1987) suggested hydrothermal circulation may play a significant role in the geochemical composition of the oceanic crust and seawater (Baker et al., 1987). Coupled fluid flow and chemical exchange probably acted to equilibrate the hydrothermal fluid with oceanic crust and modulate the chemistry of the hydrothermal field (Mottl, 2003). We proposed that the Sr and Nd isotope compositions of the Fe-Si oxyhydroxide deposits at the SWIR might be closely related to mixing between hydrothermal fluids and seawater.

RC1: The same for Sr isotopes. Your range of data is extremely variable, is not a "slight variation" going from 0.7079 to 0.7091 and you need to explain this – not done in the manuscript.

Answer: Thanks for the reviewer's constructive comments. We have moderated them in the revised paper as following:

The Sr isotopic compositions of the deposits were extremely variable (87Sr/86Sr = 0.70794–0.70915), and had values indistinguishable from present-day seawater (87Sr/86Sr = 0.70918) (Peucker-Ehrenbrink and Fiske, 2019).

RC1: Also, you are talking about oxidized systems and you quote pyrite by XRD. How abundant is the pyrite? Where it is located? Is it the primary mineral that has been extensively oxidized? Or pyrite is just a local precipitate in a more anoxic setting? The low S contents today do not prove that these rocks were originally precipitated as sulphide rocks and later being oxidized. You must have stronger arguments. These questions need to be solved by just careful observations before performing a batch of uncontrolled analytical techniques. Also, you must try to interpret all your results, even if contradictory, not just some of them.

Answer: Sorry for this mistake about the pyrite by XRD, and so at this point we have carefully reanalyzed XRD data. We think that pyrite is not determined in the hydrothermal Fe-Si oxyhydroxide deposits by XRD. We appreciate reviewer for reminding us on the discussion. We have revised the manuscript to be more clear about the XRD results as following:

The XRD results showed that 2-line ferrihydrite, hematite, nontronite, opal and birnessite composed the major minerals in the samples (Fig. S2). In the spectra of samples 21V-T7, 21V-T1 and 20V-T8, a broad peak centered at 4.08 Å suggested the presence of opal. The spectral peaks appeared at 2.69 Å and 1.60 Å in samples DIV95-1, DIV95-2, 21V-T7, 21V-T1 and 20V-T8 indicated the presence of hematite. The spectral signature of birnessite was most clearly observed in sample DIV95-2, at d = 7.06 and 2.45 Å. A small amount of birnessite was observed in DIV95-1, which was presumed to be from the residual black layer. Poorly crystalline two-line ferrihydrite, characterized by the appearance of peaks at d = 2.62 Å and 1.51 Å, was the principal mineral observed in the spectra of samples DIV95-2 and 34II-T22. Nontronite was also present in 34II-T22 deposit. In addition, halite was observed in our samples, which presumably was formed by evaporation.

RC1: Unless you unambiguously prove direct or indirectly, that the structures are microbial the major conclusions of the paper should be considered just an attractive but plausible hypothesis.

Answer: Thanks for the reviewer's constructive comments. We have added TEM images and EDS date to support the structures belong to microbial formation. Furthermore, we have also tried to culture Fe-oxidizing bacteria from these Fe-Si oxyhydroxides. We observe that Fe-stalks are produced by Fe-oxidizing bacteria similar to morphologies of Fe-Si-oxyhydroxides in studied deposits (Fig. S4). We suggest that these structures found in the iron oxydehydroxyde deposits and silica precipitates represent the microbes encrusted by Fe-Si oxydehydroxydes.

RC1: Don't go too far into speculative conclusions before being sure of that. The discussion needs to be completely rewritten but probably the aforementioned aspects need to be solved before getting into a thoughtful review.

Answer: We have deleted some too far into speculative conclusions, for example: "biogenic Fe-Si oxyhydroxides probabily tie in the origin and evolution of life" in the revised paper. We have rewritten the discussion and conclusion to enhance readability based on the reviewer's suggestions in the revised paper.

Please also note the supplement to this comment:
https://www.biogeosciences-discuss.net/bg-2019-315/bg-2019-315-AC1-supplement.pdf

———————————————————

[Figure]

**Fig. 1.** Regional bathymetric map and location of the sampling site at the SWIR. Black dots represent sample locations in this study.

[Figure]

**Fig. 2.** Chondrite-normalized REE distribution patterns of the hydrothermal Fe-Si oxyhydroxide deposits in this study.

[Figure]

**Fig. 3.** (a) Ternary diagram for Mn-Fe-(Co + Ni + Cu) ×10 of Fe-Si oxyhydroxide deposits. The hydrothermal, diagenetic and hydrogenous fields were classified by Hein et al. (1994). The Fe-Si oxyhydroxide depos

[Figure]

**Fig. 4.** SEM images showing different styles of biogenic mineral structures in different Fe-Si oxyhydroxide deposits. (a) A network-like structure composed of rod-like mineralized forms observed in the orange-

**Fig. 5.** 57Fe Mössbauer spectra at room temperature (300 K), and fitting results of Fe-Si oxyhydroxide deposits from the SWIR. (a) DIV95-1, (b) DIV95-2, (c) 34II-T22, (d) 21V-T1, (e) 21V-T7, (f) 20V-T8.

[Figure]

**Fig. 6.** Sequential extraction of iron minerals in the studied Fe-Si oxyhydroxide deposits.

[Figure]

**Fig. 7.** Comparison of 208Pb/204Pb versus 206Pb/204Pb (a), 207Pb/204Pb versus 206Pb/204Pb (b), 87Sr/86Sr versus 143Nd/144Nd (c), and 87Sr/86Sr versus 206Pb/204Pb from the studied deposits, compared against Pac

Legend: Southwest Indian Ridge Basalts; Pacific Antarctic Ridge Basalts; Arctic Ocean Sediments; Samples; Southeast Indian Ridge Basalts; Mid Atlantic Ridge Deposits; Indian Ocean Sediments; Australian-Antarctic Discordance Basalts

**Fig. 8.** Epsilon Nd versus 87Sr/86Sr for Fe-Si oxyhydroxides compared to hydrothermal fluid and seawater. Isotopic compositions of Nd and Sr based on modern Indian Ocean seawater values ($\acute{E}\check{Z}$Nd = $-8.0$, 87Sr/86Sr

[Figure]

**Fig. 9.** NanoSIMS ionic images of 12C−, 12C14N−, 32S−, 27Al16O−, 55Mn16O−, and 56Fe16O2− from a twisted stalk. Ion intensity variations were shown by calibration bars. The scale bar was 5 $\mu$m for each panel.

Table 2.   Pb, Sr, Nd,and O isotopic data for studied samples and deduced temperature.

| Sample# | $^{87}Sr/^{86}Sr$ | (2σ) | $^{143}Nd/^{144}Nd$ | (2σ) | $^{206}Pb/^{204}Pb$ | (2σ) | $^{207}Pb/^{204}Pb$ | (2σ) | $^{208}Pb/^{204}Pb$ | (2σ) | $^{147}Sm/^{144}Nd$ | εNd | $δ^{18}O$ (‰ SMOW) | Deduced temperture (°C) |
|---|---|---|---|---|---|---|---|---|---|---|---|---|---|---|
| DIV95-1 | 0.70851 | 0.000014 | 0.512441 | 0.000011 | 18.267 | 0.003 | 15.601 | 0.003 | 38.296 | 0.012 | 0.1416 | -3.8 | 20.9 | 43.2 |
| DIV95-2 | 0.70869 | 0.000013 | 0.512364 | 0.000011 | 18.307 | 0.003 | 15.595 | 0.003 | 38.331 | 0.009 | 0.1298 | -5.3 | / | / |
| 34II-T22 | 0.70915 | 0.000012 | 0.512895 | 0.000015 | 18.266 | 0.002 | 15.594 | 0.002 | 38.357 | 0.006 | 0.1180 | 5.1 | 35.8 | 114.2 |
| 21V-T1 | 0.70852 | 0.000015 | 0.512332 | 0.000015 | 18.269 | 0.002 | 15.640 | 0.003 | 38.449 | 0.006 | 0.1645 | -6 | 17.3 | 31.0 |
| 21V-T7 | 0.70794 | 0.000013 | 0.512578 | 0.000031 | 18.483 | 0.003 | 15.585 | 0.003 | 38.310 | 0.009 | 0.1924 | -1.2 | 16.5 | 28.6 |
| 20V-T8 | 0.70830 | 0.000012 | 0.512801 | 0.000015 | 18.275 | 0.001 | 15.549 | 0.001 | 38.214 | 0.003 | 0.1721 | 3.2 | 20.5 | 41.8 |

**Fig. 10.**

---

## Author Comment (AC2) · 3 Mar 2020

Response to Reviewer

All of us would like to thank the editor and reviewer again for your considerations about this manuscript and constructive comments. We have revised it very carefully. Responses are provided in a point-by-point fashion.
RC2: The paper "Formation and origin of Fe-Si oxyhydroxide deposits at the ultra-slow spreading Southwest Indian Ridge" by Ta et al. is a geochemical study of six Fe and Si rich samples from the SWIR, collected by various means between 2008 and 2015. Although the data appears well collected and analytical work is extensive and appears sound, the context of the paper, including the introduction, interpretation of results and discussion is jumbled, and there needs to be more discussion reconciling the different analytical results.

Answer: Thanks for the reviewer's constructive comments. In the revised version, we have rewritten the manuscript to enhance readability based on your suggestion. We have made considerable efforts to reconcile the different analytical results.

RC2: For example, the low sulfur content (line 436) is inconsistent with the presence of pyrite as a major mineral (line 286).

Answer: Sorry for this mistake about the pyrite by XRD. We agree with the reviewer that the low sulfur content is inconsistent with the presence of pyrite as a major mineral. We have carefully reanalyzed XRD data. We think that pyrite is not determined in the hydrothermal Fe-Si oxyhydroxide deposits by XRD. We have corrected them in the revised paper.

RC2: also none of these major minerals (line 286) represent phyllosilicates, stated as the most abundant Fe pool obtained by the leach procedure (line 340).

Answer: Thanks for the reviewer's constructive comments. We have carefully reanalyzed XRD data. Nontronite characterized by appearance peaks at d = 3.70 Å, 3.04 and 2.16 Å, was the principal phyllosilicates observed in the spectra of sample 34II-T22.

RC2: Also, the Mössbauer shows exclusively Fe(III), however XRD reveals pyrite, Fe(II). The authors claim goethite and hematite are mineral phases (line 337-338), but do not see these via XRD, The paper does not address the possible reasons for these inconsistencies, and must.

Answer: Sorry for this mistake about the pyrite by XRD, and so at this point we have carefully reanalyzed XRD data. We think that pyrite is not determined in the hydrothermal Fe-Si oxyhydroxide deposits by XRD. We appreciate reviewer for reminding us on the discussion. We have revised the manuscript to be more clear on the XRD results as following:

The XRD results showed that 2-line ferrihydrite, hematite, nontronite, opal and birnessite composed the major minerals in the samples (Fig. S2). In the spectra of samples 21V-T7, 21V-T1 and 20V-T8, a broad peak centered at 4.08 Å suggested the presence of opal. The spectral peaks appeared at 2.69 Å and 1.60 Å in samples DIV95-1, DIV95-2, 21V-T7, 21V-T1 and 20V-T8 indicated the presence of hematite. The spectral signature of birnessite was most clearly observed in sample DIV95-2, at d = 7.06 and 2.45 Å. A small amount of birnessite was observed in DIV95-1, which was presumed to be from the residual black layer. Poorly crystalline two-line ferrihydrite, characterized by the appearance of peaks at d = 2.62 Å and 1.51 Å, was the principal mineral observed in the spectra of samples DIV95-2 and 34II-T22. Nontronite was also present in 34II-T22 deposit. In addition, halite was observed in our samples, which presumably was formed by evaporation.

RC2: The discussion is also internally inconsistent. For example, regarding the role of microbial activity in forming the Fe-Si minerals line 582 suggests "these findings support the hypothesis that microbial activity was the principal deposition mechanism of Fe-Si oxyhydroxides in modern and ancient seafloor hydrothermal systems" while line 494 notes microbes "were widely involved" and line 442 suggests that they "may have played a role."

Answer: Thanks for the reviewer's constructive comments. We have moderated them in the revised paper.

RC2: The conclusions (line 584-585) and line 567-569 attempt to tie in the "origin and evolution of life" which isn't discussed in the rest of the paper and seems to be a non sequitur to the rest of the manuscript.

Answer: We agree with this comment. We have deleted this speculative conclusions in the revised paper.

RC2: Additionally, the paper should be proofread for grammatical issues and other issues, for instance there is no "Mid Pacific Ridge", the scale bars in Figure 8 are really not visible.

Answer: Thanks for the reviewer's constructive comments. In the revised version, We have made considerable efforts to improve the grammar and other issues of manuscript. We have delete "Mid Pacific Ridge". We have corrected the scale bars in Figure 9.

Please also note the supplement to this comment:
https://www.biogeosciences-discuss.net/bg-2019-315/bg-2019-315-AC2-supplement.pdf

[Figure]

**Fig. 1.** Regional bathymetric map and location of the sampling site at the SWIR. Black dots represent sample locations in this study.

[Figure]

**Fig. 2.** Chondrite-normalized REE distribution patterns of the hydrothermal Fe-Si oxyhydroxide deposits in this study.

[Figure]

**Fig. 3.** (a) Ternary diagram for Mn-Fe-(Co + Ni + Cu) ×10 of Fe-Si oxyhydroxide deposits. The hydrothermal, diagenetic and hydrogenous fields were classified by Hein et al. (1994). The Fe-Si oxyhydroxide depos

[Figure]

**Fig. 4.** SEM images showing different styles of biogenic mineral structures in different Fe-Si oxyhydroxide deposits. (a) A network-like structure composed of rod-like mineralized forms observed in the orange-

[Figure]

**Fig. 5.** 57Fe Mössbauer spectra at room temperature (300 K), and fitting results of Fe-Si oxyhydroxide deposits from the SWIR. (a) DIV95-1, (b) DIV95-2, (c) 34II-T22, (d) 21V-T1, (e) 21V-T7, (f) 20V-T8.

[Figure]

**Fig. 6.** Sequential extraction of iron minerals in the studied Fe-Si oxyhydroxide deposits.

[Figure]

**Fig. 7.** Comparison of 208Pb/204Pb versus 206Pb/204Pb (a), 207Pb/204Pb versus 206Pb/204Pb (b), 87Sr/86Sr versus 143Nd/144Nd (c), and 87Sr/86Sr versus 206Pb/204Pb from the studied deposits, compared against Pac

[Figure]

**Fig. 8.** Epsilon Nd versus 87Sr/86Sr for Fe-Si oxyhydroxides compared to hydrothermal fluid and seawater. Isotopic compositions of Nd and Sr based on modern Indian Ocean seawater values (ÉŻNd = −8.0, 87Sr/86Sr

[Figure]

**Fig. 9.** NanoSIMS ionic images of 12C−, 12C14N−, 32S−, 27Al16O−, 55Mn16O−, and 56Fe16O2− from a twisted stalk. Ion intensity variations were shown by calibration bars. The scale bar was 5 $\mu$m for each panel.

Table 2. Pb, Sr, Nd,and O isotopic data for studied samples and deduced temperature.

| Sample[a] | $^{87}Sr/^{86}Sr$ | (2σ) | $^{143}Nd/^{144}Nd$ | (2σ) | $^{206}Pb/^{204}Pb$ | (2σ) | $^{207}Pb/^{204}Pb$ | (2σ) | $^{208}Pb/^{204}Pb$ | (2σ) | $^{147}Sm/^{144}Nd$ | εNd | $δ^{18}O$ (‰ SMOW) | Deduced temperture (°C) |
|---|---|---|---|---|---|---|---|---|---|---|---|---|---|---|
| DIV95-1 | 0.70851 | 0.000014 | 0.512441 | 0.000011 | 18.267 | 0.003 | 15.601 | 0.003 | 38.296 | 0.012 | 0.1416 | -3.8 | 20.9 | 43.2 |
| DIV95-2 | 0.70869 | 0.000013 | 0.512364 | 0.000011 | 18.307 | 0.003 | 15.595 | 0.003 | 38.331 | 0.009 | 0.1298 | -5.3 | / | / |
| 34II-T22 | 0.70915 | 0.000012 | 0.512895 | 0.000015 | 18.266 | 0.002 | 15.594 | 0.002 | 38.357 | 0.006 | 0.1180 | 5.1 | 35.8 | 114.2 |
| 21V-T1 | 0.70852 | 0.000015 | 0.512332 | 0.000015 | 18.269 | 0.002 | 15.640 | 0.003 | 38.449 | 0.006 | 0.1645 | -6 | 17.3 | 31.0 |
| 21V-T7 | 0.70794 | 0.000013 | 0.512578 | 0.000031 | 18.483 | 0.003 | 15.585 | 0.003 | 38.310 | 0.009 | 0.1924 | -1.2 | 16.5 | 28.6 |
| 20V-T8 | 0.70830 | 0.000012 | 0.512801 | 0.000015 | 18.275 | 0.001 | 15.549 | 0.001 | 38.214 | 0.003 | 0.1721 | 3.2 | 20.5 | 41.8 |

**Fig. 10.**

**Supplement:**

*Supplement of*

**Formation and origin of Fe-Si oxyhydroxide deposits at the ultra-slow spreading**

**Southwest Indian Ridge**

Kaiwen Ta[1, 2], Zijun Wu[1*], Xiaotong Peng[2] and Zhaofu Luan[1]

[1]School of Ocean and Earth Science and State Key Laboratory of Marine Geology,

Tongji University, Shanghai, China.

[2]Deep Sea Science Division, Institute of Deep Sea Science and Engineering, Chinese

Academy of Sciences, Sanya, China.

Correspondence: Zijun Wu (wuzj@tongji.edu.cn)

**Contents of this file**

Supplementary Figure S1: Hydrothemal Fe-Si oxyhydroxide deposits were recovered from the ultra-slow spreading SWIR. (a) DIV95, (b) 21V-T7, (c) 21V-T1, (d) 20V-T8, (e) 34II-T22.

Supplementary Figure S2: XRD patterns of hydrothemal Fe-Si oxyhydroxide deposits at the SWIR. S1-S6 showing samples DIV95-1, DIV95-2, 34II-T22, 21V-T7, 21V-T1,

20V-T8, respectively.

Supplementary Figure S3: TEM images displaying the mineralized Fe-Si oxyhydroxides in samples 34II-T22 (a) and 20V-T8 (b). (c) EDS from the area defined by the red dot in panel a. (d) EDS from the area defined by the red dot in panel b. Cu came from Cu net in Figures c and d.

Supplementary Figure S4: (a) Fe-oxidizing bacteria gradient tube cultured with $FeS_2$.

(b) Fluorescence micrographs of cells showing filamentous morphologies (green), stained with SYBR Green I. (b) Fe-oxidizing bacteria (green).

Supplementary Table S1: Investigated hydrothermal Fe-Si oxyhydroxide deposits from the SWIR.

Supplementary Table S2: . Sequential extraction procedure of iron speciation studies and targeted minerals.

[Figure]

Supplementary Figure S1. Hydrothemal Fe-Si oxyhydroxide deposits were recovered from the ultra-slow spreading SWIR. (a) DIV95, (b) 21V-T7, (c) 21V-T1, (d) 20V-T8, (e) 34II-T22.

[Figure]

Supplementary Figure S2. XRD patterns of hydrothemal Fe-Si oxyhydroxide deposits
at the SWIR. S1-S6 showing samples DIV95-1, DIV95-2, 34II-T22, 21V-T7, 21V-T1,
20V-T8, respectively.

[Figure]

Supplementary Figure S3. TEM images displaying the mineralized Fe-Si oxyhydroxides in samples 34II-T22 (a) and 20V-T8 (b). (c) EDS from the area defined by the red dot in panel a. (d) EDS from the area defined by the red dot in panel b. Cu came from Cu net in Figures c and d.

[Figure]

Supplementary Figure S4. (a) Fe-oxidizing bacteria gradient tube cultured with FeS$_2$.

(b) Fluorescence micrographs of cells showing filamentous morphologies (green), stained with SYBR Green I. (c) Fe-oxidizing bacteria (green).

Supplementary Table S1. Investigated hydrothermal Fe-Si oxyhydroxide deposits from the SWIR.

| Sample[#] | Latitude (E) | Longitude (N) | Depth (m) | Hand sample description |
|---|---|---|---|---|
| DIV95-1 | 49.6482° | 37.7794° | 2764.3 | Orange-yellowish layer of deposits collected by Jiaolong human occupied vehicle (HOV) during the cruise of XYH09 in Feb 2015 |
| DIV95-2 | 49.6482° | 37.7794° | 2764.3 | Black layer of deposits collected by Jiaolong HOV during the cruise of XYH09 in Feb 2015 |
| 34II-T22 | 49.2580° | 37.9425° | 1499 | Purple-red deposits collected by a TV-grabber during the cruise of R/V DaYang One in Jan 2015 |
| 21V-T1 | 49.3888° | 37.4697° | 2784 | Yellowish deposits collected by a TV-grabber during the cruise of R/V DaYang One in Jan 2010 |
| 21V-T7 | 49.3894° | 37.4699° | 2746 | Brown deposits containing volcanic glass shards collected by a TV-grabber during the cruise of R/V DaYang One in Jan 2010 |
| 20V-T8 | 50.2803° | 37.3952° | 1740 | Brown deposits collected by a TV-grabber during the cruise of R/V DaYang One in Nov 2008 |

Supplementary Table S2. Sequential extraction procedure of iron speciation studies and targeted minerals.

|  | Pool | Extraction Agent | Fe Fractions |
|---|---|---|---|
| 1 | $Fe_{carb}$ | 25 mL, 1 M Na-acetate, pH 4.5, 24 h 50 °C | Carbonate iron and siderite |
| 2 | $Fe_{ox1}$ | 25 mL, 1 M hydroxylamine-HCl, 48 h | Poorly crystalline Fe (oxyhydr)oxides, ferrihydrite and lepidocrocite |
| 3 | $Fe_{ox2}$ | 25 mL, 0.28 M Na-dithionite, pH 4.8, 2 h | Goethite, hematite, and akaganeite |
| 4 | $Fe_{PRS}$ | 30mL, 12M HCl, 1 min boiling | Poorly reactive sheet silicate iron |